# Memory Type Matters: Enhancing Long-Term Memory in Large Language Models with Hybrid Strategies

## Abstract

The memory capabilities of Large Language Models (LLMs) have garnered increasing attention recently. Many approaches adopt Retrieval-Augmented Generation (RAG) techniques to alleviate the "Forgetting" problem in LLMs. Despite great success achieved, existing RAG-based memory approaches typically overlook the differences between memories and employ a unified strategy to process all memories, leading to suboptimal performance. Thus, an intuitive question arises: can we categorize memory into different types and select appropriate strategies? However, given the topic-rich, scenario-complex, and boundary-blurred nature of memory scenarios, achieving precise classification of memories is not easy. To address this challenge, we propose a memory multi-class benchmark in this paper, termed TriMEM. TriMEM comprises 6,000 dialogue samples, providing precise annotations for memory types across diverse topics and scenarios. Building upon this foundation, we propose a novel memory framework, named MemoType. MemoType can adaptively identify the category of each memory and design tailored storage and retrieval strategies, thereby achieving satisfactory performance. Extensive experiments on retrieval and generation tasks demonstrate the effectiveness of the proposed approach [1].

## 1 Introduction

Large Language Models (LLMs) have demonstrated remarkable abilities in natural language understanding and multi-turn dialogue fields, garnering increasing attention. However, with the growing frequency of User-LLM interactions, the limited context window length of LLMs fails to meet the demand for scalable conversational interactions (Liu et al., 2023). When presented with retrospective queries, LLMs cannot effectively utilize conversation memories to achieve personalized responses, resulting in the "Forgetting" problem. To address this issue, Retrieval-Augmented Generation (RAG)-based memory techniques have emerged as a practical solution. Instead of concatenating all conversations, RAG-based approaches store conversations in an external bank for selective retrieval, which enables the LLM to recall critical memories and achieve personalized responses.

RAG-based memory strategies can be broadly classified into unstructured and structured types, based on whether to construct additional memory associative information. Unstructured memory augmentation approaches typically store dialogue fragments or summaries, retrieved through similarity matching with the given query. An early and influential effort is the MemoryBank (Zhong et al., 2024), which maintains a user-specific memory and retrieves facts with relevance scoring to produce appropriate responses. On this basis, SeCom (Pan et al., 2025) employs adaptive memory segmentation techniques to achieve superior performance. Structural memory augmentation strategies organize memories into meaningful relationships, thereby achieving stronger sense-making and richer associations. For instance, HippoRAG (Gutiérrez et al.) builds a knowledge graph and uses personalized pageRank to surface associative passages. Mem0 (Chhikara et al., 2025) further advances the graph view by maintaining a dynamic memory graph that incrementally extracts entities and relations, consolidates them across dialogues. Structural memory augmentation strategies deepen memory connections while inevitably introducing additional overhead.

---

[1] https://anonymous.4open.science/r/MemoType-5578/README.md

Figure 1: Three Memory Types in TriMEM (Left) and the proposed MemoType Framework (Right).

Despite great success, the above methods still suffer certain drawbacks: i) In the retrieval phase, most approaches overlook the structural and content differences between memories and employ a unified strategy to process all memories, leading to suboptimal performance. For instance, event-related memories often contain fragmented information such as persons, time, and location, preventing regular embedding-based similarity from achieving precise retrieval. Conversely, when memories pertain only to basic concepts, introducing details like time or persons may introduce noise into the retrieval process. Thus, an intuitive question arises: *can we categorize memory into different types and select appropriate strategies?* However, given the topic-rich, scenario-complex, and boundary-blurred nature of memory scenarios, achieving precise classification of memories is not easy. Although some attempts have been conducted, most research (Wang & Chen, 2025; Zhang et al., 2025) relies on basic prompt-based methods to determine memory types, making it difficult to adapt to complex memory scenarios to achieve accurate classification, thus failing to acquire optimal results. ii) In the generation phase, most work employs retrieved memory segments directly as input to enhance LLM outputs. However, these memory segments often contain information unrelated to the query and inevitably introduce noise to the final output, resulting in performance decline.

To address these issues, we propose a memory multi-class benchmark, termed TriMEM. Inspired by cognitive psychology theory (Eysenck & Keane, 2020), three memory types are defined in TriMEM: Episodic Memory (EM), Personal Semantic Memory (PM), and General Semantic Memory (GM). Specifically, EM typically refers to a specific event, PM usually involves user information or preferences, while GM includes general knowledge and objective descriptions. TriMEM comprises 6,000 samples, providing precise annotations for memory types across diverse topics and scenarios. With the help of TriMEM, we propose a novel memory framework, MemoType, which can adaptively recognize each memory and query type with the learned router model. With the memory and query routing, MemoType can retrieve the memory with corresponding query types rather than retrieving the whole memory corpus, thereby enhancing the retrieval efficiency. Moreover, MemoType designs tailored storage and retrieval strategies for each memory type to enhance performance. Furthermore, we design a memory pruning module that adaptively prunes retrieved memories based on queries, reducing irrelevant information. Our contributions are as follows:

- Inspired by cognitive psychology theory, we propose a memory multi-class benchmark, TriMEM, which categorizes memories into three types: EM, PM, and GM, annotated across multiple topics. To the best of our knowledge, this is the first benchmark for memory type classification.

- We propose a novel memory augmentation framework, termed MemoType, which can adaptively recognize each memory and query's type and design tailored storage and retrieval strategies to enhance performance. Besides, the designed memory pruning module can adaptively prune retrieved memories based on queries, reducing interference caused by irrelevant information.

- To the best of our knowledge, MemoType is the first memory framework that adaptively classes memory and designs distinct retrieval and storage mechanisms for different memory types.

- Extensive experiments on retrieval and generation tasks show the effectiveness of MemoType.

## 2 MEMOTYPE

As emphasized in the Introduction, existing methods face challenges in both the retrieval and generation phases. In the retrieval phase, most approaches employ a uniform strategy for all memories, overlooking structural and content differences between them, leading to suboptimal performance. Besides, in the generation phase, most methods directly utilize retrieved memory segments to augment the large language models' output. However, these segments usually contain irrelevant information and introduce noise into the results, leading to performance decline. To address these issues, we propose MemoType, which consists of three modules: a memory type router module that flexibly categorises memory and query types, a hybrid strategy module that enhances performance by applying adaptive retrieval strategies to different memory types, and a memory pruning module that adaptively prunes retrieved memories based on queries to avoid noise interference.

### 2.1 PRELIMINARY

Denote $\mathcal{M} = \{\mathcal{S}_i\}_{i=1}^{S}$ as the stored conversation history between two participants, where the dialogue may occur between users or between a user and an assistant. $S$ indicates the number of sessions, $\mathcal{S}_i = \{d_l\}_{l=1}^{S_i}$ denotes the $i$-th session, comprising $S_i$ sequential dialogue. Denote the retrieval function as $f_R$ and the response generation function as $f_G$. Following the setting of SeCom (Pan et al., 2025), the pair $t_i = (d_{2i}, d_{2i+1})$ is referred to as a turn. The research framework process can be described as follows: (1) *Memory Construction*: Build a memory repository $\mathcal{B}$ from $\mathcal{M}$. (2) *Memory Retrieval*: Given a user query $q$ and a retrieval repository $B$, retrieve $n$ memory turns $\{t_i\}_{i=1}^{n} \subseteq \mathcal{B}$ relevant to $q$ using the function $f_R(q, \mathcal{B}, n)$. (3) *Response Generation*: Employ the retrieved memory turns $\{t_i\}_{i=1}^{n}$ and user query $q$ to generate the output $r$ using $f_G(q, \{t_i\}_{i=1}^{n})$.

### 2.2 MEMORY TYPE ROUTER

As mentioned in the previous sections, different types of memories require tailored strategies to optimize performance. A natural idea is to categorize memories into distinct types and apply appropriate strategies accordingly. However, due to the topic-rich, scenario-complex, and boundary-blurred nature of memory scenarios, achieving precise classification is a challenging task. While some preliminary (Wang & Chen, 2025; Zhang et al., 2025) efforts have been made in memory categorization, most existing approaches rely on basic prompt-based methods to determine memory types, which struggle to adapt to complex memory scenarios to achieve accurate classification. Experiments of Table 9 in the Appendix further demonstrate that existing Prompt-based classification methods often fail to achieve accurate categorization, limiting their effectiveness in real-world memory scenarios.

As a pioneering work in memory classification, PerLTQA (Du et al., 2024) has laid a foundation for this field. However, PerLTQA suffers from several limitations, making it less applicable to memory scenarios. First, it generates memories with single labels, failing to address conversations involving multiple topics. Second, its memory classification is incomplete: semantic memory excludes general knowledge, while episodic memory overlooks future planning. Lastly, PerLTQA adopts a coarse-grained approach, treating all dialogues as episodic memory, which lacks practicality given dialogues' significance in retrieval tasks.

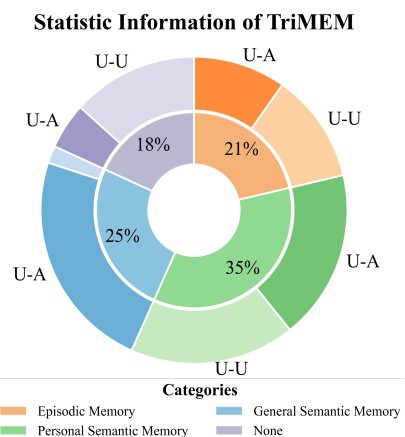

Figure 2: The Statistical Information of TriMEM[2]. "U-U" denotes the user-to-user conversation, "U-A" denotes the user-to-assistant conversation.

To tackle these challenges, we introduce a memory multi-class benchmark called TriMEM. Drawing inspiration from cognitive psychology theories (Eysenck & Keane, 2020; Du et al., 2024), this

---

[2]The percentage of the memory types is a relative ratio, as there is overlap between different memory types.

Table 1: The Memory Classification Criterion in the TriMEM Benchmark.

| Memory Type | Content Type | Example |
|---|---|---|
| EM | Event Occurred
Events Planned to Occur | "Sarah and John had dinner at a sushi restaurant downtown."
"Next Month, my colleagues are planning to see a beautiful sunset." |
| PM | User Information
User Preferences and Habits
User Intent | "Jack's degree is in Mathematics. "
"I love Italian food, especially pasta and pizza."
"I want to buy a pair of running shoes" |
| GM | General Knowledge
Broadly Applicable Suggestions
Objective Descriptions of Entities | "Water boils at 100 degrees Celsius at sea level."
"Getting up early is a good habit for maintaining good health."
"The desks in the school are made of wood." |
| None | Question without Information
Sentence without Information | "Do you have any tips on how to organize a shoe rack?"
"Good Morning, Peter"; "See you next time" |

dataset categorizes memories into three distinct types: Episodic Memory (EM), Personalized Semantic Memory (PM), and General Semantic Memory (GM). The memory classification criterion for this dataset is outlined in Table 1. Specifically, EM denotes specific events, including past events and planned events; PM covers personalized information, preferences, habits, and intentions; GM contains general knowledge facts, broadly applicable suggestions, and objective descriptions.

TriMEM comprises 6,000 dialogue instances (3,550 user-agent dialogues, 2,450 user-user dialogues). TriMEM provides memory type annotations across diverse scenarios (user-user, user-agent) and topics (life, personal information, social status, etc.). Figure 2 presents its statistical information. Further details about TriMEM and experimental results can be found in Appendix A.4.

**Memory Type Router.** With the assistance of the TriMEM benchmark, achieving precise memory classification has become significantly simplified. We utilized a basic BERT model with a three-class classifier, treating the task as a multi-label classification problem and employing binary cross-entropy with logits loss for training, requiring approximately ONLY five iterations to produce a highly accurate classifier. The classification experiment can be found in Table 9 in Appendix A.5. Compared to traditional prompt-based classification approaches, our BERT-based classification router not only achieves superior classification accuracy but also delivers lower latency and computational overhead. The training loss is shown below:

$$\mathcal{L} = -\frac{1}{N} \sum_{i=1}^{N} \sum_{c=1}^{C} \left[ y_{i,c} \cdot \log \sigma(z_{i,c}) + (1 - y_{i,c}) \cdot \log(1 - \sigma(z_{i,c})) \right], \tag{1}$$

where $N$ is the number of samples, $C$ is the classes number. $y_{i,c}$ represents the ground truth label for sample $i$ and class $c$, where $y_{i,c} \in \{0, 1\}$. $\sigma(\cdot)$ is the sigmoid activation function.

**Query Type Router.** After memory classification, another challenge is to achieve precise query routing. Using prompt-based methods for query classification also faces the issue of low accuracy. Inspired by HyDE (Gao et al., 2023)'s method of using hypothetical document generation to reduce semantic gaps between query and corpus, we instead require the LLM to generate fake memories based on the query, and then we classify these fake memories using the BERT classifier trained.

$$\{m_i\}_{i=1}^{K} = \text{LLM}_{\text{fake}}(q), \quad i = 1, \dots, K, \tag{2}$$

where $q$ is the query. The fake memory generation prompt is shown in Figure 16 in the Appendix.

During the retrieval phase, we only retrieve memories from the corpus that share the same type as the query, thereby reducing the retrieval space. Across various datasets, this strategy eliminates approximately 20% of irrelevant memory retrieval operations and reduces retrieval time. To enhance robustness, we select $K = 5$ fake memory entries and determine the core query type through a majority voting mechanism, thereby guiding the subsequent execution of the retrieval strategy. Types appearing in false memories will be identified as the retrieval scope for the query. In summary, the nature of query categories and memory categories is fundamentally different. Specifically, the query type is single, but the memory type may be multiple types or none at all, and the retrieved memory could have multiple types since nonetype memory is not in the retrieved scope of the query.

## 2.3 Hybrid Retrieval Strategy

As previously mentioned, most existing approaches employ a single strategy to process all memories, resulting in suboptimal performance. Therefore, in this subsection, we will adopt different strategies for different memory types to enhance the final performance.

**Episodic Memory.** Episodic memory usually involves events composed of multiple elements, such as time $(T)$, participants $(P)$, location $(L)$, and event context $(E)$. To address the issue of excessive coupling of key elements in traditional methods, we employ LLMs to extract event elements from both the query and memory sides. Let $\mathcal{N} = \{T, P, L, E\}$ denote the set of event element types. For a given query, the LLM extracts a subset $\mathcal{N}_{\text{query}} \subseteq \mathcal{N}$, which represents the element types in the query. The similarity score between query $q$ and the $i$-th memory is computed as:

$$s_{\text{total}}^i = \text{sim}\left(q, m^i\right) + \alpha \sum_{t \in \mathcal{N}_{\text{query}}} \text{sim}\left(e_t, m_t^i\right), \; e^t \in \mathcal{E}_{\text{query}}, m_t^i \in \mathcal{E}_{\text{memory}}^i,$$

where $\mathcal{E}_{\text{query}}$ and $\mathcal{E}_{\text{memory}}^i$ are the event element from query and the $i$-th memory, respectively. The extract prompt is shown in Figure 18. $\text{sim}(\cdot)$ is the similarity score, $\alpha$ is the parameter.

Episodic memory usually involves multi-hop retrieval and temporal reasoning, making it the most challenging among the three memory types. Although our retrieval method incurs additional overhead, the adaptive routing policy ensures that only queries involving episodic memory employ this strategy. This avoids the overhead of structure expansion for all memories seen in similar approaches, such as HippoRAG, thereby effectively reducing the latency of our method.

**Personal Semantic Memory.** The challenge of PM lies in the greater semantic gap between queries and the corpus. Queries involving personal information often result in reduced accuracy of false memories. For instance, when the query is "What is your occupation?", there are many possible answers, e.g., teacher, doctor, student, painter, which may not be exactly matched through fake memories. To mitigate this issue, we address it from the query side by utilizing LLM to generate multiple query-related keywords and concatenating them with the original query. This enhances performance without introducing additional retrieval times. The process can be formalized as:

$$q' = \text{concat}(q, \text{LLM}(q)),$$

where $q$ is the original query, $\text{LLM}(q)$ is the keywords generated by the LLM based on $q$, concat is the concatenation operation, $q'$ is the enhanced query used for retrieval.

**General Semantic Memory.** For general semantic memory, which typically involves general world knowledge and widely accepted suggestions, LLMs usually can generate high-quality responses. Therefore, no additional strategies are employed to enhance its performance. We utilize the fake memory strategy adopted for other memory types to improve overall performance. Finally, the reciprocal rank fusion (RRF) (Rackauckas, 2024) strategy is adopted to obtain the final ranking.

$$r_{\text{final}} = \text{RRF}(r, r_{fake}), \tag{3}$$

where $r$ is the rank obtained from the original query-corpus ranking or after applying the memory enhancement strategy, $r_{fake}$ is the ranking derived from the fake memory combined with the corpus.

## 2.4 Memory Pruning Module

As mentioned before, most work employs retrieved memory segments directly as input to enhance LLM outputs. However, these memory segments often contain information unrelated to the query and inevitably introduce noise to the final output, resulting in performance decline. To address this issue, we propose a memory pruning module that adaptively prunes retrieved memory based on the query, ensuring that only relevant information is retained for subsequent tasks. This module leverages the capabilities of LLMs to reason over the retrieved memory and eliminate irrelevant or noisy elements. The process can be formulated as:

$$m^p = \text{LLM}_{\text{prune}}(m^q, q), \tag{4}$$

where $m^q$ is the retrieved memory of query $q$, and the pruned memory $m^p$ would be utilized as input for the generation process. The memory pruning prompt is shown in Figure 21 in the Appendix.

## 3 EXPERIMENT

### 3.1 EXPERIMENTAL SETTINGS

**Dataset.** In the experiment part, we evaluate our model performance on three long-term memory datasets: LongMemEval-S (Wu et al., 2025), LoCoMo (Maharana et al., 2024), LongMemEval-M (Wu et al., 2025), PerLTQA (Du et al., 2024). These datasets are designed to assess the capabilities of models in both retrieval and generation tasks. Detailed dataset statistics and descriptions are provided in the Appendix B.

**Metrics.** We evaluate two types of tasks: retrieval tasks and generation tasks, using metrics tailored to each. For retrieval tasks, we use Recall@k and NDCG@k, where Recall@k measures the proportion of relevant documents retrieved in the top-k results, and NDCG@k assesses ranking quality based on relevance and position. For generation tasks, we employ BLEU, BERTScore, and GPT4Judge (Zheng et al., 2023). BLEU evaluates n-gram overlap with reference answers, BERTScore measures semantic similarity via embeddings, and GPT4Judge leverages GPT-4o to assess response alignment with reference answers. The GPT evaluation prompts are provided in Figure 23 in the Appendix.

**Baselines.** We evaluate the proposed method with various baselines. **Strong Retrieval methods**: (1) *Contriever* Izacard et al. (2021). **Three Query Expansion methods**: (2)*HyDE* (Gao et al., 2023); (3) *Mill* (Jia et al., 2023); (4) *Query2Doc* (Wang et al., 2023). **Two Memory Enhancement methods** (5) *Secom* (Pan et al., 2025); (6) *A-Mem* (Xu et al., 2025). **Two Structural RAG methods** (7) *HippoRAG 2* (Gutiérrez et al.); and (8) *RAPOTR* Sarthi et al. (2024). Detailed descriptions of baselines provided in the Appendix C.

**Implement Details.** For all methods, we adopted a consistent strategy to ensure a fair comparison. In both retrieval and generation tasks, our method and the baselines utilized Contriever as the embedding model. For generation performance, we evaluated using the top 3 retrieved turns on the LongMemEval-S and LongMemEval-M datasets, while for LoCoMo, we selected the top 10 retrieved turns for performance assessment. In our experiments, we utilize 'gpt-4o-mini' as the backbone for all tasks and baselines, including memory information processing and question-answer generation. To ensure fair comparisons, all baselines are implemented with uniform generation prompts. The temperature of the LLMs is fixed at 0 to ensure reproducibility.

### 3.2 OVERALL RESULTS

In this paper, we evaluate two types of tasks, retrieval and generation tasks, with their respective results presented in Table 2 and Table 3. Notably, RAPTOR and Mem0 involve generating a new text process, making it impossible to assess their retrieval performance. Additionally, for A-Mem, Mem0, HippoRAG2, and RAPTOR, the memory construction latency on LongMemEval-m exceeds one week, and thus, we do not report their performance in this study.

**Retrieval Results.** To evaluate the effectiveness of MemoType, we conducted retrieval performance comparison experiments, as presented in Table 2. For query expansion methods, MemoType achieves a Recall@1 of 55.74% on the LongMemEval-S dataset, outperforming Mill's by 17.02%. This highlights the limitations of traditional query expansion techniques, which struggle with personalized queries involving user-specific information. In memory enhancement, MemoType surpasses SeCom with a Recall@1 of 46.17% on the LongMemEval-M dataset, exceeding SeCom's by 14.26%. This shows that MemoType's ability to leverage memory diversity leads to superior performance compared to SeCom's simplistic segmentation. For structural RAG techniques, MemoType outperforms HippoRAG2 on the LongMemEval-S dataset, achieving a Recall@1 of 55.74% versus 50.64%, a 5.1% improvement. This suggests that structural RAG methods may overcomplicate tasks, while MemoType's approach better balances simplicity and effectiveness, adapting to query complexity and memory characteristics.

**Question Answer Results.** To validate the question answer performance of the proposed method, we conduct the QA experiment in Table 3. For query expansion methods, our approach demonstrated strong results, particularly on BLEU and BERTScore. On the LongMemEval-S dataset, it achieved a BLEU score of 4.52, far surpassing Query2Doc and Mill, highlighting the limitations of traditional methods in generating high-quality, contextually relevant answers. For memory enhance-

Table 2: Retrieval Performance. All methods are based on Contriever as the retriever.

| Method | Recall@1 | NDCG@1 | Recall@3 | NDCG@3 | Recall@5 | NDCG@5 | Recall@10 | NDCG@10 |
|---|---|---|---|---|---|---|---|---|
| LongMemEval-S | | | | | | | | |
| Contriever | 43.40 | 43.40 | 73.62 | 54.00 | 83.83 | 60.33 | 94.26 | 65.79 |
| HyDE | 27.23 | 27.23 | 53.19 | 37.82 | 65.32 | 42.84 | 82.34 | 49.83 |
| Mill | 38.72 | 38.72 | 67.66 | 50.17 | 78.94 | 56.10 | 90.64 | 61.34 |
| Query2Doc | 25.96 | 25.96 | 50.64 | 36.43 | 64.89 | 41.66 | 81.06 | 48.38 |
| SeCom | 45.32 | 45.32 | 74.04 | 53.42 | 83.19 | 59.42 | 91.28 | 64.66 |
| A-Mem | 27.02 | 27.02 | 44.89 | 29.54 | 52.98 | 33.17 | 67.02 | 37.64 |
| HippoRAG2 | 50.64 | 50.64 | 80.85 | 60.59 | 88.51 | 66.48 | 93.53 | 71.05 |
| Ours | **55.74** | **55.74** | **81.06** | **63.40** | **88.72** | **68.64** | **93.62** | **72.54** |
| LoCoMo | | | | | | | | |
| Contriever | 20.75 | 20.75 | 36.56 | 30.86 | 44.06 | 34.16 | 53.93 | 37.31 |
| HyDE | 25.33 | 25.33 | 42.55 | 36.24 | 50.45 | 39.60 | 59.82 | 42.65 |
| Mill | 19.84 | 19.84 | 35.60 | 30.16 | 42.75 | 33.10 | 53.12 | 36.57 |
| Query2Doc | 23.11 | 23.11 | 39.78 | 33.94 | 47.48 | 37.15 | 59.37 | 41.11 |
| SeCom | 23.77 | 23.77 | 39.38 | 34.06 | 45.57 | 36.60 | 55.44 | 39.76 |
| A-Mem | 11.03 | 11.03 | 17.93 | 14.96 | 21.85 | 16.47 | 28.50 | 18.42 |
| HippoRAG2 | 24.97 | 24.97 | 39.98 | 33.84 | 45.87 | 36.28 | 53.17 | 38.64 |
| Ours | **26.54** | **26.54** | **43.15** | **36.77** | **50.60** | **39.79** | **61.48** | **43.56** |
| LongMemEval-M | | | | | | | | |
| Contriever | 33.19 | 33.19 | 57.87 | 41.04 | 68.09 | 46.46 | 82.98 | 51.77 |
| HyDE | 21.70 | 21.70 | 40.43 | 27.90 | 50.64 | 31.99 | 63.62 | 36.71 |
| Mill | 28.51 | 28.51 | 52.77 | 37.48 | 63.40 | 42.73 | 76.81 | 48.04 |
| Query2Doc | 19.15 | 19.15 | 37.02 | 25.66 | 44.68 | 28.96 | 61.49 | 34.03 |
| SeCom | 31.91 | 31.91 | 55.11 | 38.16 | 64.47 | 42.38 | 75.96 | 47.06 |
| Ours | **46.17** | **46.17** | **71.91** | **52.95** | **80.21** | **58.35** | **87.45** | **62.29** |
| PerLTQA | | | | | | | | |
| HyDE | 17.09 | 17.09 | 56.06 | 53.16 | 66.42 | 58.01 | 79.49 | 62.52 |
| Mill | 51.87 | 51.87 | 77.66 | 75.29 | 84.36 | 78.47 | 91.40 | 80.92 |
| Query2Doc | 26.15 | 26.15 | 62.26 | 59.34 | 72.90 | 64.36 | 83.47 | 68.04 |
| A-Mem | 25.11 | 25.11 | 30.47 | 39.57 | 45.50 | 41.91 | 59.77 | 56.67 |
| HippoRAG2 | 41.47 | 41.47 | 72.23 | 70.02 | 78.37 | 72.91 | 85.80 | 75.50 |
| Ours | **59.48** | **59.48** | **80.39** | **78.69** | **86.39** | **81.05** | **92.47** | **82.83** |

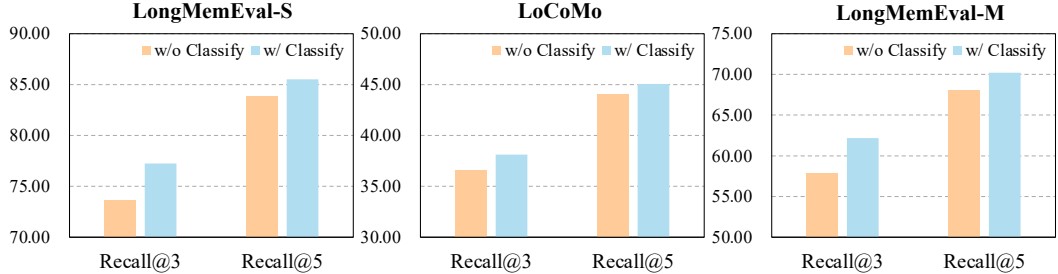

Figure 3: The ablation study of the proposed classification strategy on three datasets. "w/ Classify" represents adopting the classification strategy to filter memory, "w/o Classify" denotes the opposite.

ment techniques, our method excelled on LoCoMo, achieving the highest GPT4Judge score of 40.38 and a BERTScore of 85.43, outperforming approaches like SeCom that fail to utilize memory diversity effectively. For structural RAG methods, our approach showed clear advantages by avoiding the inefficiencies of overcomplicated structural enhancements, demonstrating that selectively applying techniques based on query complexity is critical for optimal performance. These results underscore the robustness of our method.

Table 3: Question Answer Performance. All methods are based on Contriever as the retriever.

| Method | GPT4Judge | F1 | BLEU | Rouge1 | Rouge2 | RougeL | RougeLsum | BERTScore |
|---|---|---|---|---|---|---|---|---|
| LongMemEval-S | | | | | | | | |
| Contriever | 44.40 | 10.40 | 1.60 | 11.05 | 4.89 | 9.79 | 10.07 | 83.06 |
| HyDE | 42.60 | 9.96 | 1.44 | 10.58 | 4.67 | 9.31 | 9.57 | 83.02 |
| Mill | 42.20 | 9.91 | 1.55 | 10.57 | 4.54 | 9.20 | 9.45 | 82.96 |
| Query2Doc | 43.00 | 9.80 | 1.47 | 10.44 | 4.46 | 9.21 | 9.43 | 82.97 |
| SeCom | 44.80 | 11.01 | 1.65 | 11.66 | 5.14 | 10.36 | 10.58 | 83.26 |
| A-Mem | 27.80 | 8.51 | 1.27 | 9.15 | 3.57 | 7.74 | 7.99 | 82.52 |
| HippoRAG2 | 45.40 | 10.55 | 1.65 | 11.15 | 5.17 | 9.98 | 10.17 | 83.12 |
| Raptor | 32.20 | 12.08 | 1.90 | 12.73 | 5.82 | 11.25 | 11.35 | 83.50 |
| Ours | **50.00** | **20.27** | **4.52** | **21.13** | **10.49** | **19.54** | **19.71** | **85.32** |
| LoCoMo | | | | | | | | |
| Contriever | 37.97 | 14.54 | 2.44 | 14.97 | 7.21 | 13.98 | 13.97 | 84.38 |
| HyDE | 40.18 | 14.86 | 2.60 | 15.25 | 7.35 | 14.14 | 14.15 | 84.48 |
| Mill | 37.41 | 14.50 | 2.49 | 14.88 | 7.13 | 13.84 | 13.83 | 84.38 |
| Query2Doc | 39.83 | 14.57 | 2.50 | 14.95 | 7.16 | 13.88 | 13.87 | 84.44 |
| SeCom | 38.57 | 14.85 | 2.41 | 15.30 | 7.37 | 14.25 | 14.25 | 84.44 |
| A-Mem | 25.03 | 11.14 | 1.63 | 11.75 | 5.12 | 10.93 | 10.90 | 83.73 |
| HippoRAG2 | 37.81 | 14.61 | 2.56 | 15.03 | 7.33 | 14.03 | 14.04 | 84.42 |
| Raptor | 31.72 | 14.55 | 2.88 | 15.09 | 7.49 | 14.18 | 14.17 | 84.48 |
| Ours | **40.38** | **19.69** | **4.53** | **20.10** | **10.08** | **18.91** | **18.94** | **85.43** |
| LongMemEval-M | | | | | | | | |
| Contriever | 35.40 | 9.20 | 1.34 | 9.87 | 4.15 | 8.59 | 8.85 | 82.79 |
| HyDE | 34.60 | 9.01 | 1.26 | 9.71 | 3.99 | 8.46 | 8.67 | 82.70 |
| Mill | 35.40 | 8.79 | 1.29 | 9.47 | 3.89 | 8.14 | 8.39 | 82.58 |
| Query2Doc | 34.80 | 8.66 | 1.20 | 9.34 | 3.88 | 7.99 | 8.23 | 82.61 |
| SeCom | 34.40 | 10.13 | 1.37 | 10.83 | 4.65 | 9.51 | 9.69 | 83.07 |
| Ours | **41.60** | **17.79** | **3.78** | **18.73** | **8.72** | **17.14** | **17.29** | **85.02** |
| PerLTQA | | | | | | | | |
| HyDE | 39.39 | 30.80 | 7.21 | 32.59 | 16.41 | 26.82 | 26.90 | 88.86 |
| Mill | 51.37 | 36.65 | 10.61 | 38.41 | 21.40 | 32.44 | 32.50 | 89.85 |
| Query2Doc | 42.32 | 31.95 | 7.82 | 33.72 | 17.46 | 27.95 | 28.01 | 89.06 |
| A-Mem | 32.12 | 34.96 | 5.20 | 29.95 | 19.64 | 31.94 | 22.01 | 87.74 |
| HippoRAG2 | 49.26 | 34.44 | 8.89 | 36.24 | 20.01 | 30.48 | 30.57 | 89.42 |
| Ours | **52.85** | **42.49** | **17.17** | **44.32** | **26.49** | **38.69** | **38.66** | **90.92** |

## 3.3 ABLATION STUDY

In this section, we conduct an ablation study to analyze the effectiveness of our proposed method from four key perspectives: classification strategy, hybrid retrieval strategy, memory pruning strategy, and the design of the retriever. The following subsections detail the experimental settings and results for each aspect, offering a thorough examination of the robustness of our method.

**Classification Strategy.** As shown in Figure 3, the proposed classification strategy effectively improves retrieval accuracy. In the experiment, "w/Classify" indicates filtering the memory corpus using our query labels; "w/o Classify" indicates retrieving all memories without category filtering. No retrieval enhancement techniques were employed in this experiment to eliminate interference from other strategies. As shown in Table 3, after introducing memory-type routing, retrieval performance across all metrics consistently improved, indicating that queries were correctly routed to their corresponding memories. For instance, on LongMemEval-S, Recall@5 exceeds 90%, and on LoCoMo and LongMemEval-M, recall improves significantly. Moreover, for the more challenging recall@1 and recall@3 metrics, the improvement in retrieval precision is most significant, indicating that the router effectively filters out some interfering memories through type-based filtering. More detailed experiment results can be shown in Appendix D.

Table 4: The ablation study of the proposed hybrid strategy on three datasets. "Key" and "Fake Memory" denote the enhancement of retrieval using keywords and fake memory, respectively.

| Strategy | LongMemEval-S | | | LoCoMo | | | LongMemEval-M | | |
|---|---|---|---|---|---|---|---|---|---|
| | Recall@1 | Recall@3 | Recall@5 | Recall@1 | Recall@3 | Recall@5 | Recall@1 | Recall@3 | Recall@5 |
| Native | 43.40 | 73.62 | 83.83 | 20.75 | 36.56 | 44.06 | 33.19 | 57.87 | 68.09 |
| Key | 43.82 | 75.10 | 84.25 | 20.90 | 37.06 | 44.61 | 33.19 | 60.63 | 66.59 |
| Fake Memory | 51.06 | 78.72 | 87.44 | 23.23 | 42.40 | 49.48 | 41.48 | 69.36 | 78.72 |
| Our Strategy | **55.74** | **81.06** | **88.72** | **26.54** | **43.15** | **50.60** | **46.17** | **71.91** | **80.21** |

Table 5: The ablation study of memory pruning strategy on three datasets. "w/ Prun" denotes adopting the proposed memory pruning strategy, and"w/o Prun" denotes the contrary.

| Strategy | LongMemEval-S | | | LoCoMo | | | LongMemEval-M | | |
|---|---|---|---|---|---|---|---|---|---|
| | GPT4Judge | BLEU | BERTScore | GPT4Judge | BLEU | BERTScore | GPT4Judge | BLEU | BERTScore |
| w/o Prun | 49.00 | 3.15 | 82.88 | 39.88 | 4.18 | 85.19 | **42.00** | 2.80 | 84.44 |
| w/ Prun | **50.00** | **4.52** | **85.32** | **40.38** | **4.53** | **85.43** | 41.60 | **3.78** | **85.02** |

**Hybrid Retrieval Strategy.** To validate the hybrid retrieval strategy in improving retrieval performance, we conducted experiments as shown in Table 4. We compared this hybrid strategy with a single strategy (all memory types employ the same retrieval-enhancing strategy) and investigated its impact on retrieval performance. "Key" indicates the enhancement of retrieval using keywords, while "Fake Memory" represents the use of fake memory for improvement. On the LongMemEval-S dataset, our strategy achieved a Recall@1 of 55.74, surpassing the "Fake Memory" approach by 4.68 points. Similarly, on LoCoMo and LongMemEval-M datasets, our strategy outperformed others in Recall@3 and Recall@5. These results confirm that the hybrid retrieval strategy effectively leverages the unique advantages of different memory types to further boost performance.

**Memory Pruning Strategy.** As shown in Table 5, our method significantly improves QA performance across all datasets. In the experiments, "w/o Prun" represents no use of the memory pruning strategy, while "w/ Prun" indicates its application. On the LongMemEval-S dataset, our method achieves higher BLEU (4.52) and BERTScore (85.32) compared to "w/o Prun." Similar improvements are observed on LoCoMo and LongMemEval-M. These results demonstrate that our memory pruning strategy effectively filters irrelevant information, enhancing retrieval and QA accuracy.

**Different Retriever.** As shown in Table 6, the ablation study evaluates different retrievers combined with our method across three datasets. For all base retrievers (MPNet, MiniLM, and QAMiniLM), our approach consistently achieves the highest recall scores at ranks 1, 3, and 5. For instance, with MPNet on LongMemEval-S, our method achieves 38.72 (Recall@1) and 77.45 (Recall@5), significantly outperforming other methods. Similar trends are observed with MiniLM and QAMiniLM across all datasets. These results demonstrate that our method effectively enhances retrieval performance by leveraging base retrievers and optimizing memory utilization for improved recall accuracy.

# 4 RELATED WORK

**Cognitive Psychology.** In cognitive psychology (Eysenck & Keane, 2020), memory is commonly divided into different types, with episodic and semantic memory being key components of declarative memory. Episodic memory involves recalling specific events (Anokhin et al., 2024), including details like time, people, places, and emotions tied to the experience. In contrast, semantic memory encompasses general knowledge about the world, such as facts, concepts, and meanings, which are not tied to specific events. Episodic memory retrieval benefits from contextual cues, but irrelevant temporal or spatial information can hinder retrieval for non-episodic queries. Unlike generalized semantic memory, personalized semantic memories often involve a larger gap between the query and corpus, requiring tailored retrieval strategies. Categorizing memory types is crucial for advancing memory research and improving retrieval performance.

**RAG based Memory technology** RAG-based memory strategies can be broadly classified into unstructured and structured types. Unstructured RAG (Lu et al., 2023; Zhang et al., 2025; Yang et al., 2025b; Zhong et al., 2024; Chhikara et al., 2025) stores history as dense chunks or LLM summaries

Table 6: The ablation study of different retrievers.

| Method | LongMemEval-S | | | LoCoMo | | | LongMemEval-M | | |
|---|---|---|---|---|---|---|---|---|---|
| | Recall@1 | Recall@3 | Recall@5 | Recall@1 | Recall@3 | Recall@5 | Recall@1 | Recall@3 | Recall@5 |
| Base Retriever: MPNet | | | | | | | | | |
| MPNet | 26.38 | 50.85 | 63.62 | 19.13 | 34.24 | 41.89 | 18.51 | 36.17 | 44.04 |
| HyDE | 25.11 | 53.19 | 66.60 | 27.29 | 44.76 | 53.52 | 17.87 | 38.09 | 46.81 |
| Mill | 27.45 | 53.19 | 64.89 | 21.00 | 36.81 | 44.51 | 19.15 | 39.36 | 48.30 |
| Query2Doc | 27.45 | 54.68 | 68.72 | 26.38 | 46.12 | 54.03 | 19.36 | 40.43 | 49.36 |
| SeCom | 37.66 | 62.13 | 73.19 | 21.45 | 34.24 | 41.49 | 26.81 | 40.64 | 48.72 |
| Ours | **38.72** | **66.17** | **77.45** | **29.36** | **48.49** | **56.80** | **27.45** | **49.36** | **60.64** |
| Base Retriever: MiniLM | | | | | | | | | |
| MiniLM | 35.74 | 63.40 | 74.04 | 14.20 | 28.50 | 35.60 | 27.45 | 47.23 | 57.87 |
| HyDE | 32.34 | 61.70 | 72.55 | 22.00 | 37.92 | 45.87 | 22.34 | 44.89 | 54.47 |
| Mill | 36.81 | 63.62 | 75.11 | 16.26 | 31.27 | 38.97 | 26.81 | 47.45 | 57.45 |
| Query2Doc | 29.79 | 61.49 | 72.34 | 22.91 | 38.32 | 46.78 | 21.28 | 41.70 | 51.70 |
| SeCom | 37.02 | 64.04 | 77.45 | 16.11 | 28.54 | 34.74 | 28.94 | 47.45 | 57.23 |
| Ours | **45.32** | **74.04** | **82.98** | **26.03** | **43.05** | **51.01** | **35.11** | **60.21** | **68.51** |
| Base Retriever: QAMiniLM | | | | | | | | | |
| QAMiniLM | 35.32 | 61.70 | 73.62 | 11.83 | 22.00 | 27.59 | 27.02 | 46.38 | 58.09 |
| HyDE | 31.91 | 61.06 | 73.19 | 17.27 | 30.56 | 37.61 | 21.70 | 43.40 | 54.68 |
| Mill | 35.32 | 63.19 | 74.04 | 9.42 | 18.98 | 26.13 | 25.53 | 48.51 | 57.02 |
| Query2Doc | 33.62 | 61.06 | 75.11 | 19.08 | 32.63 | 40.94 | 24.47 | 47.02 | 56.60 |
| SeCom | 41.06 | 71.06 | 81.28 | 15.56 | 28.80 | 36.60 | **32.77** | **53.40** | 61.91 |
| Ours | **42.34** | **74.04** | **82.97** | **23.51** | **40.79** | **48.34** | 31.49 | **58.30** | **66.38** |

and retrieves by similarity; representative systems include TiM's post-think memories (Liu et al., 2023), and SeCom's segment-level denoising (Pan et al., 2025). Structured RAG (Gutiérrez et al.; Edge et al., 2024; He et al., 2024; Jin et al., 2024; Guo et al., 2024) introduces relations via graphs or trees: Zep's (Rasmussen et al., 2025) Graphiti integrates conversations and business data into a temporal KG, MemTree (Rezazadeh et al., 2024) organizes a dynamic tree, and H-MEM (Sun & Zeng, 2025). Structural information augmentation can improve performance on complex reasoning tasks, but it often incurs inevitable computational overhead. For simpler problems, such enhancements may introduce noise and degrade performance. Our proposed Memotype addresses this issue effectively by categorizing memory into different types and adopting tailored strategies for each.

**Agent Memory** Beyond RAG, agents learn or structure memory with richer mechanisms (Wang et al., 2024; Wang et al.; Wang & Chen, 2025; Kang et al., 2025). MemoryLLM (Wang et al., 2024) updates parameters with a latent memory pool. Memory-R1 (Yan et al., 2025) trains a memory manager and answer agent with RL to add/update/delete entries. Learn-to-Memorize models memory cycles with MoE-gated retrieval, aggregation, and reflection; $Mem^p$ (Fang et al., 2025) distills procedural traces into lifelong skills. Through methods such as reinforcement learning, these approaches often achieve superior performance compared to traditional RAG systems. However, they typically incur higher memory construction latency. Balancing performance and latency in real-world scenarios remains an area worthy of further exploration for these methodologies. Moreover, considering that non-declarative memory (Eysenck & Keane, 2020) corresponds to the agent memory. Thus, exploring valuable classifications of non-declarative memory, alongside designing appropriate strategies for each type, is still a promising direction for advancing agent memory systems.

## 5 CONCLUSION

In this paper, we propose TriMEM, a multi-class memory benchmark inspired by cognitive psychology, categorizing memory into Episodic Memory, Personalized Semantic Memory, and General Semantic Memory. Built on this foundation, we introduce MemoType, a novel memory framework that adaptively identifies memory types and designs tailored storage and retrieval strategies, significantly enhancing performance. Extensive experiments demonstrate the effectiveness of this approach. This research explored distinctions between different memory types from the perspective of query expansion. Whether different memory types at the index and corpus levels can benefit from further optimisation strategies remains a highly promising direction for future research.

ETHICS STATEMENT

This work adheres to the ICLR Code of Ethics. In this study, no human subjects or animal experimentation were involved. All datasets used, including LongMemEval-S, LoCoMo, and LongMemEval-M, were sourced in compliance with relevant usage guidelines, ensuring no violation of privacy. We have taken care to avoid any biases or discriminatory outcomes in our research process. No personally identifiable information was used, and no experiments were conducted that could raise privacy or security concerns. We are committed to maintaining transparency and integrity throughout the research process.

REPRODUCIBILITY STATEMENT

We have made every effort to ensure that the results presented in this paper are reproducible. All code and datasets have been made publicly available in an anonymous repository[3] to facilitate replication and verification. The experimental setup, including training steps, model configurations, and hardware details, is described in detail in the paper.

Additionally, all data used in this paper, including LongMemEval-S, LoCoMo, and LongMemEval-M, are publicly available, ensuring consistent and reproducible evaluation results.

We believe these measures will enable other researchers to reproduce our work and further advance the field.

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

## A BENCHMARK

### A.1 CLASSIFICATION BASIS

In this paper, we will focus on text classification in the form of dialogue. Because dialogues are crucial retrieval objects in existing memory scenarios. For instance, in the two common memory benchmarks LoCoMo (Maharana et al., 2024) and LongMemEval (Wu et al., 2025), dialogues are the primary and exclusive retrieval objects. Moreover, in real-world scenarios, accessible data typically exists in dialogue form. We propose three memory types: Episodic Memory, Personal Semantic Memory, and General Semantic Memory. This classification is rooted in cognitive psychology research on declarative memory (Eysenck & Keane, 2020), which encompasses memory for facts and events that can be consciously recalled and linguistically described. In this subsection, we will explain the basis for our classification from the perspectives of completeness and necessity. Specifically, we will demonstrate that the three categories of memory encompass all conversational memory and justify the necessity of this classification.

**Completeness.** Our classification is based on well-established research in cognitive psychology (Eysenck & Keane, 2020) on declarative memory. Declarative memory inherently includes both semantic memory and episodic memory, which collectively cover all forms of dialogue memory that can be expressed through language. Nondeclarative memory, such as procedural memory or priming memory, cannot be articulated linguistically and thus lies beyond the scope of this framework. Therefore, the three proposed categories comprehensively encompass all dialogue memory types without requiring additional classifications.

**Necessity.** The necessity of categorizing semantic memory into Personal Semantic Memory and General Semantic Memory is grounded in two critical considerations. First, the division addresses the category imbalance problem, as semantic memory accounts for a disproportionately large proportion (87.64%) of dialogue memory compared to episodic memory (12.35%). Subdividing semantic memory ensures a more balanced classification, mitigating the dominance of one category during model optimization and evaluation. Second, the semantic gap between Personal and General Semantic Memory is empirically validated through TSNE visualization and word cloud analysis. As shown in the Figure 4, both categories exhibit distinct clustering patterns, indicating significant semantic differences between personal

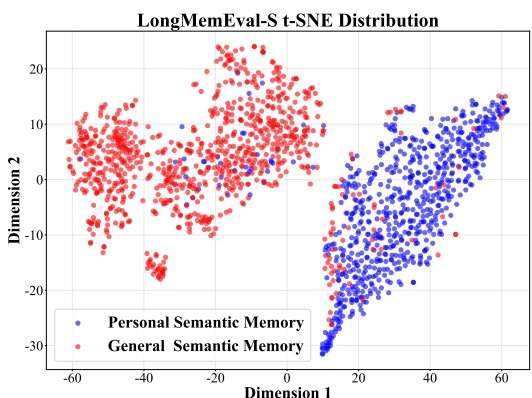

Figure 4: The t-SNE Distribution Visualization of LongMemEval-S.

and general semantic memory. Additionally, we supplemented PM Word Cloud Visualizations and GM Word Cloud Visualizations as shown in Figure 5 and Figure 6, revealing that General Semantic Memory emphasizes common terms like "assistant" and "offer," while Personal Semantic Memory focuses on user-specific terminology such as "planning "and "user." This further highlights the distinctions between the two memory types. This clear semantic distinction underscores the necessity of separating semantic memory into these two subcategories.

### A.2 CLASSIFICATION INDUCTION

In this subsection, we offer a more comprehensive explanation of the three types of memory.

- **Episodic Memory.** Episodic memory refers to memories of specific events or occurrences tied to contextual details such as time, location, participants, and activities. These memories can include both past experiences and plans for the future. For example: "Last summer, I attended a three-day conference in Paris where I presented my research on renewable energy." "Next Friday, I'm scheduled to meet my supervisor at 10 a.m. to review the final draft of my thesis."
- **Personal Semantic Memory.** Personal Semantic Memory refers to long-term knowledge related to an individual's identity, preferences, habits, or intentions. Unlike Episodic Memory, it focuses

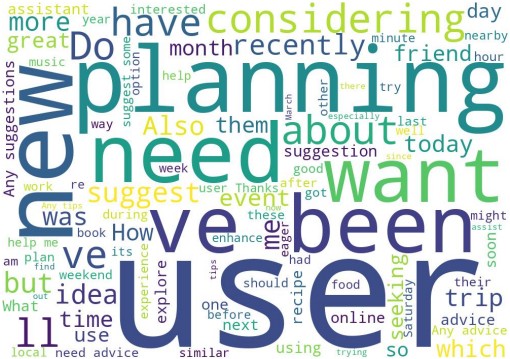

Figure 5: The Word Cloud Figure of Personal Semantic Memory on LongMemEval-S Dataset.

Figure 6: The Word Cloud Figure of General Semantic Memory on LongMemEval-S Dataset.

on general personal information rather than specific events. For instance: "I have been working as a data scientist for a long time, and I specialize in natural language processing." "I usually prefer hiking in the mountains over beach vacations because I enjoy the tranquility of nature."

- **General Semantic Memory.** General Semantic Memory encompasses universally applicable knowledge, objective descriptions, or facts about the world. It is not tied to any specific individual or event. Examples include: "The Great Wall of China is over 13,000 miles long and was constructed to protect against invasions." "Photosynthesis is the process by which green plants convert sunlight into chemical energy stored in glucose."

## A.3 BENCHMARK CONSTRUCTION

TriMEM is a carefully curated dataset comprising 6,000 dialogue samples, designed to evaluate long-term memory capabilities in dialogue models. The dataset is constructed with diversity and quality in mind, and its creation involves multiple steps including sampling, rewriting, and labeling, all of which are detailed below.

**Data Sampling.** TriMEM is composed of dialogue samples sourced from multiple datasets and manually constructed dialogues to ensure a balanced and realistic testbed. *LongMemEval-S* (Wu et al., 2025): 1,000 dialogue pairs were sampled from this dataset, which contains approximately 20,000 dialogue records. To prevent over-representation, the sampling proportion was kept low at only 10%. *LoCoMo* (Maharana et al., 2024): 750 dialogue pairs were sampled to add variety. *PersonaMem* (Jiang et al., 2025): 750 dialogue pairs were also sampled to further diversify the dataset. *Manually-constructed dialogues*: 500 dialogue pairs were created by human annotators to introduce additional variability and ensure coverage of edge cases not captured in the sampled datasets.

**Benchmark Selection.** We aimed to annotate memory categories in real-world dialogue scenarios rather than generating memory-related dialogues from predefined categories, which would risk introducing bias by over-focusing on single categories. We chose LoCoMo (Maharana et al., 2024) and LongMemEval (Wu et al., 2025) as they are widely recognized memory datasets frequently referenced in works such as A-Mem (Xu et al., 2025), mem0 (Chhikara et al., 2025), MIXIR (Wang & Chen, 2025), and Secom (Pan et al., 2025). This highlights their significance and credibility. LongMemEval emphasizes user-assistant interactions, while LoCoMo focuses on user-user dialogues, aligning with our goal of covering diverse memory scenarios and topics. To further enhance the generalizability of our annotated dataset beyond evaluation-specific categories, we included PersonaMem (Jiang et al., 2025), a personalized memory benchmark that introduces varied user profiles. This addition strengthens the dataset's applicability across broader dialogue classification tasks.

**Rewriting to Prevent Overfitting.** Since both LongMemEval-S and LoCoMo are also used in the evaluation datasets of this study, special care was taken to avoid the risk of the model directly memorizing specific dialogue samples. Such memorization could compromise the evaluation of classification performance.

---

**Transfer Prompt for User-Assistant Conversation**

You are an advanced text transformation AI. Your task is to process a given conversation between a user and an assistant. Follow these rules:

- Replace specific nouns (e.g., occupations, professional fields, locations, dates, names, times, and objects) with alternative but relevant ones, ensuring the topic of the conversation remain unchanged.
- Summarize both the user's and assistant's messages.
- Limit user's responses to 40 words.
- Limit assistant's responses to 80 words.
- You can rearrange the order of the sentences to maintain better coherence between them.

**Output Format:** Output the revised conversation in this format:
[user]: [content]
[assistant]: [content]

**Input**: {text}

---

Figure 7: Transfer Prompt for User-Assistant Conversation

---

**Transfer Prompt for User-User Conversation**

You are an advanced text transformation AI. Your task is to process a given conversation between users. Follow these rules:

- Replace specific nouns (e.g., names, occupations, professional fields, locations, dates, times, and objects) with alternative but relevant ones, ensuring the topic of the conversation remain unchanged.
- You may try rearranging the order of the sentences to maintain better coherence between them.

**Output Format:** Output the revised conversation in this format:
[user name 1]: [content]
[user name 2]: [content]

**Input**: {text}

---

Figure 8: Transfer Prompt for User-User Conversation

To address this, the sampled dialogues from these datasets were rewritten using transfer prompt, which is shown in Figure 7 and Figuer 8. The rewriting process involved modifying the necessary entity names, rephrasing relevant topic statements, and substituting key details while preserving the original semantic structure. This ensured that the dialogues remained contextually meaningful but sufficiently distinct from their original forms, mitigating the risks of data leakage and overfitting.

**Labeling Process.** The labeling of dialogue samples was conducted through a combination of automated and manual processes to ensure high accuracy and consistency. *Initial Labeling*: The GPT-4o-2024-11-20 model was used to generate initial labels for the dialogue samples. These prompts were meticulously crafted and validated on a subset of the data to ensure their effectiveness for the classification task. Notably, we opted for separate prompts for each category, as this approach achieved higher classification accuracy compared to using specific prompts for all categories (Episodic Memory Classification Prompt in Figure 9, Personal Semantic Memory Classification Prompt in Figure 10, and General Semantic Memory Classification Prompt in Figure 11). These prompts were tailored to the unique characteristics of each memory category, ensuring precise and context-aware labeling. *Manual Validation and Refinement*: Recognizing the potential for errors in automated labeling, all

Table 7: The Statistic Information of TriMEM.

| Statistic | Value | Ratio (%) |
|---|---|---|
| # Dialogue (User–Assistant) | 3,550 | 59.17 |
| # Dialogue (User-User) | 2,450 | 40.83 |
| # Total Dialogue | 6,000 | 100.00 |
| Average tokens per text | 88.26 | N/A |

initial labels were thoroughly reviewed and corrected by human annotators. This combination of automated and manual processes guaranteed the reliability and accuracy of the annotations.

By integrating diverse data sources, robust rewriting, and automated/manual labeling, TriMEM ensures a high-quality, diverse, and practical dataset. This makes it a reliable benchmark for evaluating long-term memory in dialogue models, minimizing risks of overfitting and data leakage biases.

### A.4 BENCHMARK STATISTIC INFORMATION

The TriMEM dataset comprises 6,000 dialogues, with a nearly 60:40 split between User–Assistant (59.17%) and User–User (40.83%) interactions, as shown in Table 7. Each text contains an average of 88.26 tokens, highlighting its moderate linguistic complexity.

As can be seen in Table 8, class-wise, the dataset demonstrates diverse distribution across four categories: Episodic Memory (26.07%), Personal Semantic Memory (43.18%), General Semantic Memory (30.75%), and None (22.03%). Notably, user–assistant dialogues dominate the General Semantic Memory class (92.73%), while user–user dialogues contribute significantly to the None class (73.68%). Meanwhile, the Episodic and Personal Semantic Memory classes show a more balanced distribution across interaction types, with slight variations favoring user–assistant dialogues.

This distribution reflects the dataset's richness and its potential to model different memory types in varied conversational contexts. By capturing both assistant-guided and peer-to-peer interactions, TriMEM offers a comprehensive resource for advancing memory-based dialogue systems, supporting tasks like memory reasoning and personalized conversation.

### A.5 BENCHMARK EVALUATION

In this study, we evaluated the classification performance of four large language models (LLMs) on the TriMEM dataset: Qwen3-8b (Yang et al., 2025a), Gemini-2.5-Flash (Comanici et al., 2025), GPT-4o-Mini (Hurst et al., 2024), and Qwen3-32b with multi-class prompt in Figure 12. Additionally, our model was included for comparison. The evaluation metrics used were accuracy (ACC) and F1 score, which respectively measure the overall classification correctness and the balance between precision and recall. Performance was analyzed across three memory dimensions in TriMEM: Episodic Memory (EM), Personal Semantic Memory (PM), and General Semantic Memory (GM).

Experimental results indicate that our model outperformed all other LLMs in every category. For overall accuracy and F1 score, our model achieved 92.10% and 91.75%, respectively, significantly surpassing GPT-4o-Mini, the second-best model, which scored 83.55% (ACC) and 76.94% (F1). In the EM dimension, our model achieved an outstanding 98.80% (ACC) and 97.81% (F1), outperforming GPT-4o-Mini, which achieved 85.05% (ACC) and 76.10% (F1). For PM, our model reached 82.65% (ACC) and 85.64% (F1), while GPT-4o-Mini achieved slightly higher accuracy (87.18%) but lower F1 (85.59%). Finally, in GM, our model achieved 94.85% (ACC) and 92.39% (F1), outperforming others by a clear margin.

Despite our model's superior performance, the results highlight that existing LLMs struggle to achieve precise classification on TriMEM, requiring further advancements in memory modeling for improved accuracy and generalization.

## B DATASET DETAILS

We evaluate our model with the following datasets:

---

**Episodic Memory Classification Prompt**

**Goal**: You are an advanced AI tasked with determining whether a specific piece of dialogue contains Episodic Memory. Use the following definition, guidelines, and examples to identify whether the provided dialogue qualifies as Episodic Memory.

**Definition**:
Episodic Memory refers to specific episodes or events that occurred (are planned to occur, or the thoughts on future plans) at a particular place and time. For the dialogue to qualify as Episodic Memory, it often include details:

- When: A time, date, time frame, or vague contextual time expressions (e.g., "yesterday," "next month", "when I was driving").
- Where: A location or context (e.g., "at the park," "in Paris," "at the office").
- Who: Participants (e.g., "with my classmates," "Peter").
- What: Actions, or objects involved in the event (e.g., "attending a party," "have lunch").

**Additional Guidelines:**

- The time and location information does not always need to be precise. Vague or contextual expressions such as "when I was driving," "during lunch", or "while I was in school," can also qualify as time-related or location-related details if they provide sufficient information for the event.
- Questions that reveal or imply specific episodes or events information, should also be classified as Episodic Memory.
- Missing one types of detail does not disqualify the dialogue from being Episodic Memory if the other details are sufficiently explicit to describe a instance.
- If the memory only describes general habits, vague knowledge, or patterns, it does not qualify as Episodic Memory.
- If the dialogue contains at least one instance of Episodic Memory, the output must be "Episodic Memory", regardless of whether other parts of the dialogue do not contain Episodic Memory.

**Output Requirements:**

- If the dialogue contains Episodic Memory, output: "Episodic Memory"
- If the dialogue does not contain Episodic Memory, output: "None"
- Do not include additional context, comments, symbols, or explanations in your output

**Input**: $\{text\_to\_be\_classified\}$

**Output**: Now, based on the above instructions, determine whether the input data contains Episodic Memory.

Figure 9: Episodic Memory Classification Prompt

Table 8: The Class Distribution of TriMEM.

| Class | Percentage (%) | User–Assistant (%) | User-User (%) |
|---|---|---|---|
| Episodic Memory | 26.07 | 45.83 | 54.17 |
| Personal Semantic Memory | 43.18 | 50.66 | 49.34 |
| General Semantic Memory | 30.75 | 92.73 | 7.27 |
| None | 22.03 | 26.32 | 73.68 |

- **LongMemEval-S** (Wu et al., 2025) is a dataset designed to evaluate long-term memory in conversational AI systems. It includes 50 question sessions, each with an average of 115,000 tokens, providing a compact yet challenging benchmark. This dataset tests core memory abili-

---

**Personal Semantic Memory Classification Prompt**

**Goal**: You are an advanced AI tasked with determining whether a segment of dialogue contains Personal Semantic Memory. Use the following definition, guidelines, and examples to identify whether the provided dialogue qualifies as Personal Semantic Memory.

**Definition**:
Personal Semantic Memory is a form of long-term memory consisting of knowledge specifically related to the person (user or another individual mentioned in the dialogue), such as descriptions of their information, preferences, habits, thoughts, background, emotions, education or work career-related information, and any other knowledge that reflects the individual's identity.

**Additional Guidelines:**

- Content that reveals personal information, preferences, habits, plans, background information, emotions, personal history, education or work career-related information, or any other knowledge that related to the individual's identity must be classified as Personal Semantic Memory.

- Statements that express the individual's intentions, work arrangements, or preliminary plans (not include details), even if they are inferred, belong to the category of Personal Semantic Memory.

- General knowledge unrelated to personal information, preferences, or emotions do not qualify as Personal Semantic Memory.

- Questions that reveal or imply personal details, preferences, or habits (user or another individual), should also be classified as Personal Semantic Memory.

- If the dialogue contains at least one instance of Personal Semantic Memory, the output must be "Personal Semantic Memory", regardless of whether other parts of the dialogue do not contain Personal Semantic Memory. The unanswered questions or requests in the conversation should not affect the final classification result.

**Output Requirements:**

- If the dialogue contains Personal Semantic Memory, output: "Personal Semantic Memory"

- If the dialogue does not contain Personal Semantic Memory, output: "None"

- Do not include additional context, comments, symbols, or explanations in your output

**Input**: $\{text\_to\_be\_classified\}$

**Output**: Now, based on the above instructions, determine whether the input data contains Personal Semantic Memory.

---

Figure 10: Personal Semantic Memory Classification Prompt

ties such as information extraction, multi-session reasoning, temporal reasoning, and abstention. LongMemEval-s requires chat assistants to process user-AI dialogues effectively, retain pertinent information over extended conversation histories, and maintain consistency across sessions. By focusing on shorter yet complex contexts, LongMemEval-s serves as a foundational benchmark for assessing memory performance.

- **LoCoMo** (Maharana et al., 2024) is a long-context memory benchmark, evaluates AI systems' ability to handle lengthy dialogues and long-range dependencies. It features conversations with an average of 300 turns, 9,000 tokens, and up to 35 sessions, with a publicly available subset, LoCoMo-10, comprising ten high-quality conversations. The dataset is well-suited for testing long-context language models (LLMs) and retrieval-augmented generation (RAG) systems. De-

---

**General Semantic Memory Classification Prompt**

**Goal**: You are an advanced AI tasked with determining whether a given text contains General Semantic Memory based on the following definition, criteria, and examples.

**Definition**:
General Semantic Memory is a form of long-term memory consisting of non-personalized knowledge about the world, including facts, concepts, language, universally applicable methods, and objective descriptions of locations or objects. This includes information that is not tied to any specific individual or event.
For information to qualify as General Semantic Memory, it must meet at least one of the following criteria:

- Non-Personalized Knowledge or Broadly Applicable Suggestions: It could include general knowledge, factual information, or widely applicable suggestions and methods. Examples: Scientific facts, historical events, cultural concepts, general knowledges, or general advice on common activities.

- Objective Descriptions of Locations, Objects, or Concepts: Information that provides objective, factual details about locations, objects, or entities. Examples: Descriptions of landmarks, places, or any items.

**Additional Guidelines:**

- Focus on analyzing sentences that provide valid information. Dialogue that consists solely of questions without answers should not be used as a basis for General Semantic Memory classification.

- The information about a specific event or personalized to the user (e.g., user preferences, personal habits, or individual-specific details) does not qualify as General Semantic Memory.

- If the dialogue contains at least one instance of General Semantic Memory, the output must be "General Semantic Memory", regardless of whether other parts of the dialogue do not contain General Semantic Memory.

**Output Requirements:**

- If the dialogue contains General Semantic Memory, output: "General Semantic Memory"

- If the dialogue does not contain General Semantic Memory, output: "None"

- Do not include additional context, comments, symbols, or explanations in your output

**Input**: $\{text\_to\_be\_classified\}$

**Output**: Now, based on the above instructions, determine whether the input data contains General Semantic Memory.

Figure 11: General Semantic Memory Classification Prompt

spite advances, systems still struggle with temporal reasoning and maintaining coherence in extended interactions, highlighting the challenge of replicating human-like memory performance.

- **LongMemEval-M** (Wu et al., 2025) builds on LongMemEval-S with a more extensive evaluation setting, encompassing 500 sessions per question and an average of 1.5 million tokens per conversation. This dataset is designed for rigorous testing of long-term memory and scalability. It challenges systems to handle dynamic user-AI interactions across an even greater number of sessions, requiring robust memory mechanisms for historical consistency and information recall. The dataset underscores the substantial performance gap between state-of-the-art AI systems and human capabilities in long-term memory retention and reasoning over extended contexts.

Table 9: The classification Performance on TriMEM with four LLM models. "EM" denotes Episodic Memory, "PM" denotes the Personal Semantic Memory, "GM" denotes the General Semantic Memory.

| Model | Overall | | EM | | PM | | GM | |
|---|---|---|---|---|---|---|---|---|
| | ACC | F1 | ACC | F1 | ACC | F1 | ACC | F1 |
| Qwen3-8b | 71.90 | 55.26 | 81.65 | 49.05 | 71.37 | 55.58 | 62.68 | 61.15 |
| Gemini-2.5-Flash | 77.84 | 72.75 | 82.47 | 73.47 | 76.72 | 74.55 | 74.33 | 70.22 |
| GPT-4o-Mini | 83.55 | 76.94 | 85.05 | 76.10 | **87.18** | 85.59 | 78.42 | 69.14 |
| Qwen3-32b | 77.75 | 72.03 | 87.82 | 76.80 | 80.88 | 75.67 | 64.55 | 63.61 |
| Ours | **92.10** | **91.75** | **98.80** | **97.81** | 82.65 | **85.64** | **94.85** | **92.39** |

## C  BASELINE DETAILS

We evaluate our model with the following baselines:

- **Contriever** (Izacard et al., 2021) is a retrieval model that serves as a strong baseline by efficiently retrieving and ranking relevant documents based on dense representations.

- **MPNet** (Song et al., 2020) is a pre-trained transformer model that combines masked language modeling and permuted language modeling to better capture dependencies between tokens for improved natural language understanding.

- **MiniLM** (Wang et al., 2020) is a lightweight and efficient transformer-based model designed for natural language understanding tasks, offering competitive performance with significantly fewer parameters compared to larger models.

- **QAMiniLM** (Wang et al., 2020) is an extension of MiniLM, designed specifically for Question Answering tasks, leveraging MiniLM's lightweight architecture while fine-tuning it to optimize performance for extracting precise answers from text.

- **HyDE** (Gao et al., 2023) generates a "hypothetical" document using a language model and retrieves similar real documents via dense embeddings to improve zero-shot dense retrieval without needing labeled relevance data. The prompt of HyDE could be found in Figure 13.

- **Mill** (Jia et al., 2023) utilizes large language models to generate diverse sub-queries and documents, followed by a mutual verification process to synergize generated and retrieved data for effective zero-shot query expansion. The prompt of Mill could be found in Figure 14.

- **Query2Doc** (Wang et al., 2023) improves sparse and dense retrieval systems by generating pseudo-documents through few-shot prompting of large language models (LLMs) and using them for query expansion, achieving significant performance boosts without fine-tuning. The prompt of QueryDoc could be found in Figure 15.

- **Secom** (Pan et al., 2025) enhances memory retrieval in long-term conversations by segmenting them into topically coherent units and applying compression-based denoising for improved retrieval accuracy and semantic quality.

- **A-Mem** (Xu et al., 2025) introduces an agentic memory system for LLMs that dynamically organizes and evolves memories by leveraging Zettelkasten principles, enabling adaptive and context-aware knowledge management through dynamic linking and indexing.

- **HippoRAG2** (Gutiérrez et al.)  is a framework that enhances retrieval-augmented generation (RAG) by integrating deeper passage connections and effective LLM use, achieving superior performance on factual, sense-making, and associative memory tasks, thereby advancing non-parametric continual learning for LLMs.

- **Raptor** (Sarthi et al., 2024) improves retrieval-augmented language models by recursively embedding, clustering, and summarizing text into a hierarchical tree, enabling multi-level abstraction retrieval and achieving state-of-the-art performance in complex reasoning tasks.

---

**Memory Multi-Classification Prompt**

**Goal**: You are an advanced AI tasked with classifying a dialogue between a user and a chatbot into one or more of the following memory types: "Episodic Memory," "Personal Semantic Memory," or "General Semantic Memory." Use the following definitions to guide your classification:

- Episodic Memory: General Semantic Memory is a form of long-term memory consisting of non-personalized knowledge about the world, including facts, concepts, language, universally applicable methods, and objective descriptions of locations or objects. This includes information that is not tied to any specific individual or event.

- Personal Semantic Memory: Personal Semantic Memory is a form of long-term memory consisting of knowledge specifically related to the person (user or another individual mentioned in the dialogue), such as descriptions of their information, preferences, habits, thoughts, background, emotions, education or work career-related information, and any other knowledge that reflects the individual's identity.

- General Semantic Memory: A form of long-term memory consisting of general knowledge about the world, including facts, concepts, language, universally applicable methods, and objective description. This includes information that is broadly relevant, commonly recognized, or widely recommended.

**Output Requirements:**
Your output must strictly adhere to one or more of the following memory types, separated by commas if multiple apply:

- "Episodic Memory"
- "Personal Semantic Memory"
- "General Semantic Memory"

If none of the above types apply, the output must be:

- "None"

Do not include any additional words, or context beyond the specified output format. Carefully analyze the dialogue and determine the appropriate memory type(s) based on the content and context provided.

**Input**: $\{text\_to\_be\_classified\}$

**Output**: Now, classify the memory type(s) of the input data based on the instructions above.

Figure 12: Memory Multi-Classification Prompt

---

**HyDE Prompt**

Please write a paragraph that answers the question.
**Question:** $\{query\}$
**Output:**

Figure 13: HyDE Prompt

# D ADDITIONAL EXPERIMENTS

## D.1 STRATEGY CONTRIBUTIONS OF EACH TYPE

To investigate the independent contribution of each memory strategy , we supplemented the study with ablation experiments for each strategy category, as shown in Table 10. Each row of data repre-

---

**MILL Prompt**

What sub-queries should be searched to answer the following query?
Please generate 5 sub-queries with their related passages.
**Question:** {query}
Only present the subquestion. without any other words and explanation.

---

Figure 14: MILL Prompt

---

**Query2Doc Prompt**

Write a passage that answers the given query:
**Question:** $\{query\}$
**Output:**

---

Figure 15: Query2Doc Prompt

sents the effect of applying only one strategy (GM, PM, or EM) of one specific memory type. The results in Table 10 show that each memory type strategy independently contributes to performance improvement, with the EM strategy yielding the most significant gains .

### D.2 THE ABLATION STUDY OF BERT ENCODER

In this section, we performed ablation studies to evaluate the impact of using alternative encoders, such as RoBERTa and DeBERTa. As evidenced in Table 11, both RoBERTa and DeBERTa achieved competitive results after fine-tuning on the TriMEM dataset, indicating that our dataset is not restricted to a specific model architecture. Notably, RoBERTa delivered better classification performance compared to the baseline BERT model, highlighting its superior modeling capacity.

### D.3 CLASSIFICATION STRATEGY ABLATION STUDY.

To verify the effectiveness of our proposed classification strategy in improving memory filtering, we conducted experiments on three datasets, as shown in Table 12. Here, "w/ Classify" represents using the classification strategy, while "w/o Classify" indicates its absence. Notably, in the LongMemEval-S dataset, "w/ Classify" achieved a Recall@1 of 46.17%, surpassing "w/o Classify" by 2.77 percentage points. Similarly, in the LoCoMo dataset, Recall@3 improved by 1.55 points. In LongMemEval-M, "w/ Classify" consistently outperformed in metrics like NDCG@10. These results demonstrate that the classification strategy significantly enhances performance across multiple datasets.

### D.4 OOD CLASSIFICATION

In this subsection, we will discuss scenarios where memory and queries are assigned to nonetype in our classification scenario and its implications.

"NoneType" memories typically correspond to questions with no information, such as "What is your name?" or generic statements like "Hello," "Good morning," or "Nice to meet you." These kinds of texts not only increase retrieval latency but may also interfere with correctly retrieving relevant content from the corpus. Therefore, when constructing the TriMEM dataset, we intentionally included NoneType type to filter out such meaningless inputs and improve retrieval performance.

Regarding NoneType queries, as previously stated, the query type is conditional on the type that occurs most frequently among the generated fake memories. Since queries in practical applications are often meaningful, such as "What is John's major?" and fake memories should contain information to answer the given query, it is rare for LLMs to generate fake memories containing meaningless

> **Answer Generation Prompt for User-Assistant Conversation**
>
> **Goal**: Suppose a user has recent conversation records with assistant. Use your imagination to generate conversation records. The generated conversation records must contain information related to the given query. The conversation must be logically clear and structurally reasonable, representing a discussion on a specific topic rather than a direct recollection of the query.
> Conversation record should be generated from one of the following perspectives:
>
> - When the query is not an advice-seeking type of question, a sentence can be output that directly provides an answer matching the query, avoiding expressions other than stating the answer.
>
> - Instead of providing a direct answer, it can describe the user's preferences, habits, events, or background related to the topic of the query. This is not a direct answer to the query, but should be an additional statement from the user, especially for advice-seeking type queries.
>
> **Input Data**: The input is a natural language query posed by the user, typically related to the previous conversations information.
>
> **Output Requirements:**
>
> - The output must not exceed one sentence. You should determine whether the query-related information is mentioned by the user or the assistant.
>
> - Ensure that the response contains only the dialogue content of one speaker.
>
> **Input**: $\{query\_to\_be\_answered\}$
>
> **Output**: Generate a hypothetical dialogue record.

Figure 16: Answer Generation Prompt for User-Assistant Conversation

Table 10: Strategy Contribution of each type on three datasets.

| Method | Recall@1 | NDCG@1 | Recall@3 | NDCG@3 | Recall@5 | NDCG@5 | Recall@10 | NDCG@10 |
|---|---|---|---|---|---|---|---|---|
| | | | | LongMemEval-S | | | | |
| Native | 46.17 | 46.17 | 77.23 | 57.51 | 85.53 | 63.43 | 94.04 | 68.46 |
| with GM Strategy | 47.45 (+1.28) | 47.45 (+1.28) | 77.87 (+0.64) | 58.16 (+0.65) | 85.53 (+0.00) | 63.81 (+0.38) | 93.92 (-0.12) | 68.68 (+0.22) |
| with PM Strategy | 47.87 (+1.70) | 47.87 (+1.70) | 77.28 (+0.05) | 58.26 (+0.75) | 86.81 (+1.28) | 64.71 (+1.28) | 94.17 (+0.13) | 69.49 (+1.03) |
| with EM Strategy | 52.77 (+6.60) | 52.77 (+6.60) | 80.28 (+3.05) | 62.02 (+4.51) | 87.45 (+1.92) | 66.98 (+3.55) | 93.92 (-0.12) | 71.30 (+2.84) |
| | | | | LoCoMo | | | | |
| Native | 21.9 | 21.9 | 38.11 | 32.17 | 45.06 | 35.11 | 54.83 | 38.24 |
| with GM Strategy | 22.00 (+0.10) | 22.00 (+0.10) | 38.12 (+0.01) | 32.22 (+0.05) | 45.12 (+0.06) | 35.15 (+0.04) | 54.88 (+0.05) | 38.30 (+0.06) |
| with PM Strategy | 23.51 (+1.61) | 23.51 (+1.61) | 40.68 (+2.57) | 34.33 (+2.16) | 47.89 (+2.83) | 37.22 (+2.11) | 57.00 (+2.17) | 40.39 (+2.15) |
| with EM Strategy | 24.82 (+2.92) | 24.82 (+2.92) | 40.58 (+2.47) | 34.57 (+2.40) | 47.73 (+2.67) | 37.64 (+2.53) | 59.26 (+4.43) | 41.37 (+3.13) |
| | | | | LongMemEval-M | | | | |
| Native | 35.53 | 35.53 | 62.17 | 44.47 | 70.21 | 49.14 | 82.98 | 54.48 |
| with GM Strategy | 36.81 (+1.28) | 36.81 (+1.28) | 62.55 (+0.38) | 44.75 (+0.28) | 71.06 (+0.85) | 49.66 (+0.52) | 83.19 (+0.21) | 54.80 (+0.32) |
| with PM Strategy | 37.02 (+1.49) | 37.02 (+1.49) | 64.47 (+2.30) | 46.47 (+2.00) | 73.40 (+3.19) | 51.94 (+2.80) | 84.26 (+1.28) | 56.55 (+2.07) |
| with EM Strategy | 43.40 (+7.87) | 43.40 (+7.87) | 69.15 (+6.98) | 50.67 (+6.20) | 76.17 (+5.96) | 55.04 (+5.90) | 85.96 (+2.98) | 59.91 (+5.43) |

responses like "Hello" or counter-questions like "What is your name?", which is why NoneType queries did not occur in our experiments.

# E   LLM USAGE DISCLOSURE

Large Language Models (LLMs) were used to aid in the writing and polishing of the manuscript. Specifically, we used an LLM to assist in refining the language, improving readability, and ensuring

**Answer Generation Prompt for User-User Conversation**

**Goal**: Suppose there is a conversation between users. Imagine you are the user mentioned in the query and generate a dialogue in the first person. The generated dialogue record must contain relevant information that answers the given query. The memory must be logical, well-structured, and tailored to the type of question provided.

**Input Data**: The input is a query in natural language that asks a specific question or seeks information.

**Output Requirements:**

- Completeness: Ensure that each dialogue memory provides sufficient information to address the query comprehensively.

- Length: The output must not exceed one to two sentences. Ensure brevity while maintaining clarity and relevance.

- Perspective: The response should be written from the perspective of the user mentioned in the query, as a single speaker.

**Output Format:**
[user name in query]: [response content]

**Input**: {$query\_to\_be\_answered$}

**Output**: Generate a hypothetical dialogue record.

Figure 17: Answer Generation Prompt for User-User Conversation

**Query Event Extract Prompt**

**Instruction**: Identify and extract the elements of events mentioned in the query.

**Output Requirements:**
Use the following structured format for output:

- Time: [Time or 'N/A']

- Person(s): [Person(s) involved or 'N/A']

- Location: [Location or 'N/A']

- Event: [Event description]

**Additional guidance**
When processing the query, always treat the subject "I" as a person and include it under the "Person(s)" field in the output.

**Input**: {$text\_to\_be\_processed$}

Figure 18: Query Event Extract Prompt

clarity in various sections of the paper. The model helped with tasks such as sentence rephrasing, grammar checking, and enhancing the overall flow of the text.

It is important to note that the LLM was not involved in the ideation, research methodology, or experimental design. All research concepts, ideas, and analyses were developed and conducted by the authors. The contributions of the LLM were solely focused on improving the linguistic quality of the paper, with no involvement in the scientific content or data analysis.

---

**Conversation Event Extract Prompt**

**Instruction**: Identify and extract the elements of events mentioned in the conversation.
**Output Requirements:**
Use the following structured format for output:

- [Event n]

- Time: [Time or 'N/A']

- Person(s): [Person(s) involved or 'N/A']

- Location: [Location or 'N/A']

**Additional guidance**

- replace all occurrences of "User" in the Person(s) field with "I".

- If multiple events are mentioned, repeat the structure above, separating each event block with a blank line.

**Input**: $\{text\_to\_be\_processed\}$

---

Figure 19: Conversation Event Extract Prompt

---

**Conversation Keywords Expansion Prompt**

**Instruction**: Please provide additional search keywords for each of the key aspects of the following queries that make it easier to find the relevant documents. Do not include irrelevant text and separate the search topics with commas.
**Input**: Query is: $\{query\}$

---

Figure 20: Conversation Keywords Expansion Prompt

The authors take full responsibility for the content of the manuscript, including any text generated or polished by the LLM. We have ensured that the LLM-generated text adheres to ethical guidelines and does not contribute to plagiarism or scientific misconduct.

---

**Memory Pruning Prompt**

**Role:** You are a memory extractor.
**Goal:** Given a Query and a Memory block, output the exact substrings from Memory that are relevant to the Query. Do not rewrite, paraphrase, translate, summarize, comment, or add any characters that are not present in Content.
**Strict rules:**

- Output only characters that literally appear in Content. Preserve original order, casing, punctuation, whitespace, speaker, and line breaks.

- Include ALL relevant substrings. Do not omit any relevant line even if other lines also seem sufficient.

- If an image caption contains information relevant to the query, it should be included in the final output.

- Speaker tags: Preserve the exact leading speaker label format "[xx]:", unchanged. Keep timestamps that appear in the same corpus if present, unchanged.

**Silence rule:**

- If no substring is relevant to the Query, output nothing (empty response).

- If multiple relevant substrings are disjoint, output them concatenated in their original order with no extra characters inserted.

**Input format:**
Query: {question}
**Memory:** {context}
**Procedure:**

- Scan the entire Content line by line.

- Speaker/Caption tags: Preserve the exact leading speaker label format "[speaker name]:".

---

Figure 21: Memory Pruning Prompt

---

**Response Prompt**

You are an intelligent dialogue bot. You will be shown Related Evidences supporting for User Input, and Recent Dialogs between user and you. Please read, memorize, and understand given materials, then generate one concise, coherent and helpful response. Provide the answer itself directly, without including any other statements.
{context}
**Question**: {question}

---

Figure 22: Response Prompt, which follows (Lu et al., 2023; Pan et al., 2025)

> **GPT Judge Prompt**
>
> I will give you a question, a reference answer, and a response from a model. Please answer [[yes]] if the response contains the reference answer. Otherwise, answer [[no]]. If the response is equivalent to the correct answer or contains all the intermediate steps to get the reference answer, you should also answer [[yes]]. If the response only contains a subset of the information required by the answer, answer [[no]].
> [User Question] $\{question\}$
> [The Start of Reference Answer] $\{answer\}$ [The End of Reference Answer]
> [The Start of Model's Response] $\{response\}$ [The End of Model's Response]
> Is the model response correct? Answer [[yes]] or [[no]] only.

Figure 23: GPT Judge Prompt, which follows (Wu et al., 2025; Zheng et al., 2023)

Table 11: The Ablation Study of BERT Encoder.

| Method | Recall@1 | NDCG@1 | Recall@3 | NDCG@3 | Recall@5 | NDCG@5 | Recall@10 | NDCG@10 |
|---|---|---|---|---|---|---|---|---|
| | | | | LongMemEval-S | | | | |
| BERT | 55.74 | 55.74 | 81.06 | 63.40 | 88.72 | 68.64 | 93.62 | 72.54 |
| DeBERTa | 55.32 | 55.32 | 81.91 | 63.19 | 89.36 | 68.90 | 95.74 | 72.62 |
| RoBERTa | 58.09 | 58.09 | 83.83 | 65.27 | 90.21 | 70.70 | 95.53 | 74.41 |
| | | | | LoCoMo | | | | |
| BERT | 26.54 | 26.54 | 43.15 | 36.77 | 50.60 | 39.79 | 61.48 | 43.56 |
| DeBERTa | 26.74 | 26.74 | 44.16 | 38.08 | 51.71 | 41.15 | 62.08 | 44.56 |
| RoBERTa | 27.49 | 27.49 | 43.96 | 37.46 | 51.31 | 40.52 | 62.13 | 44.27 |
| | | | | LongMemEval-M | | | | |
| BERT | 46.17 | 46.17 | 71.91 | 52.95 | 80.21 | 58.35 | 87.45 | 62.29 |
| DeBERTa | 46.38 | 46.38 | 68.94 | 49.73 | 77.66 | 55.55 | 86.81 | 59.91 |
| RoBERTa | 49.15 | 49.15 | 69.57 | 51.96 | 79.79 | 57.52 | 88.51 | 62.12 |

Table 12: The ablation study of the proposed classification strategy on three datasets. "w/ Classify" represents adopting the classification strategy to filter memory, "w/o Classify" denotes the opposite.

| Method | Recall@1 | NDCG@1 | Recall@3 | NDCG@3 | Recall@5 | NDCG@5 | Recall@10 | NDCG@10 |
|---|---|---|---|---|---|---|---|---|
| | | | | LongMemEval-S | | | | |
| w/o Classify | 43.40 | 43.40 | 73.62 | 54.00 | 83.83 | 60.33 | 94.26 | 65.79 |
| w/ Classify | **46.17** | **46.17** | **77.23** | **57.51** | **85.53** | **63.43** | 94.04 | **68.46** |
| | | | | LoCoMo | | | | |
| w/o Classify | 20.75 | 20.75 | 36.56 | 30.86 | 44.06 | 34.16 | 53.93 | 37.31 |
| w/ Classify | **21.9** | **21.9** | **38.11** | **32.17** | **45.06** | **35.11** | **54.83** | **38.24** |
| | | | | LongMemEval-M | | | | |
| w/o Classify | 33.19 | 33.19 | 57.87 | 41.04 | 68.09 | 46.46 | 82.98 | 51.77 |
| w/ Classify | **35.53** | **35.53** | **62.17** | **44.47** | **70.21** | **49.14** | 82.98 | **54.48** |

