# OpenReview forum: "Memory Type Matters: Enhancing Long-Term Memory in Large Language Models with Hybrid Strategies"
_ICLR.cc/2026/Conference — Submitted to ICLR 2026_

### Official Review · Reviewer_ipaj · 2025-10-15

**Soundness:** 3
**Presentation:** 2
**Contribution:** 2
**Rating:** 4
**Confidence:** 3

**Summary:**

Inspired by psychology, the paper proposes to categorize memories into three types: episodic memory, personal semantic memory and general semantic memory and use these memory types to tailor retriever in the RAG system to improve the RAG performance. The paper has compared its designed RAG system with an extensive number of SOTA RAG systems on challenging RAG benchmarks of involving long memory context and show favorable performance. The improvement is shown for the retrieval performance as well as the RAG answer performance using corresponding metrics.

The ablation studies in the paper show that the proposed memory classification indeed improves the overall retrieval performance compared to the RAG system without such classifications. The retrieval strategy like "fake memory" and using keywords are effective for the retrieval performance and that the proposed strategy holds improvement across various existing retrievers.

**Strengths:**

- The paper is well motivated in its writing and the proposed methodology is quite clear for the readers to understand including its experimental settings as well as the motivations.
- The paper has compared with an extensive number of SOTA rag systems with different focus (e..g structured RAG, memory enhancement, query enhancement) and the experimental results show the effectiveness of the proposed methods.
- Such proposal seems new. Besides, the implementation is simple and this potentially means the paper can have a good impact in its adoption in real life scenarios. The novelty itself comes with a dataset and annotation which is a contribution on its own and can benefit the community by its introduction.

**Weaknesses:**

Given that the paper's main contribution is the in introduction of memory classification and the ensuing improvement, there is lack in its detailed explanations and ablations. I will leave the explanations in the question section. But I would be curious to see: what is the effect of each retrieval since they are designed separately for each memory type (maybe even more detailed as the classification is multi-label, raising question of the effectiveness for the overlapping part)? What justifies the keyword method adoption in personal semantic memory?
Without answering these detailed questions, I don't have a clear idea how and why the proposed memory enhance the RAG performance, particularly the improvement in each domain.

I do agree that the ablation is not trivial to construct including what baseline to choose to answer these questions; nevertheless, I think this question corresponds to the main theme and the contribution of the paper.

I don't get a clear idea how the classifier will go for the OOD classification (does not fall into the classified types).

**Questions:**

- Is fake memory technology used for personal semantic memory as well?
- How does the trained classifier handle and be trained for the classes that do not fall into existing categories?
- In ablation study (classify), is the reported performance without classify but with key and fake strategies?
- In ablation study (Hybrid), is keyword only impacting the personal semantic memory?
- In related works agent memory, it seems that what the paper proposes can naturally enhance the existing agent memory techniques by letting agent possess/learn different strategies based on memory type. Do authors agree and have further thoughts to share on this?

---

> ### Author Response · Authors · 2025-11-21
> **Response to Reviewer ipaj [Part 1]**
>
> **Dear Reviewer ipaj**:
>
> Thank you so much for your thoughtful review and valuable feedback. We deeply appreciate the time and effort you devoted to evaluating our work. Below, we have provided detailed responses to your comments.
>
> ## **1. Single Retrieval Strategy Ablation**
>
> > **W1:** what is the effect of each retrieval since they are designed separately for each memory type (maybe even more detailed as the classification is multi-label, raising question of the effectiveness for the overlapping part)?
>
> Thanks for your thoughtful question. We apologize for any confusion caused by the original paper. To address your concerns, we have conducted additional ablation experiments on three datasets to evaluate the effectiveness of single retrieval strategies, as shown in Table 1. **The experiment and related statement have been added in `Table 10` on Page 24 and  `Appendix D.1` on Page 22 in the revised version.**
>
> - **The experimental results in Table 1 demonstrate that almost all retrieval strategies improve retrieval performance compared to the native baseline**. Among these, the strategies tailored for episodic memory (EM Strategy) show the most significant improvement across all metrics, suggesting that the episodic memory retrieval mechanism has the greatest impact on enhancing retrieval performance.
> - We also want to clarify a potential misunderstanding: the query type is determined by the type of the most frequent label among the five "fake memories" generated by the LLM. As such, **the query itself is not multi-labeled**. However, **the retrieved memories may belong to multiple types or none at all**. For each query, the retrieval strategy is singular and fixed, but the memories retrieved may span multiple categories.
>
> **Table 1: The ablation study of a single retrieval strategy on three datasets.** GM, PM, EM represent General Semantic Memory,  Personal Semantic Memory, and Episodic Memory, respectively.
>
> | Method  | Recall@1  | NDCG@1  | Recall@3  | NDCG@3   | Recall@5          | NDCG@5            | Recall@10         | NDCG@10           |
> | ----------------- | ----------------- | ----------------- | ----------------- | ----------------- | ----------------- | ----------------- | ----------------- | ----------------- |
> | **LongMemEval-S** |                   |                   |                   |                   |                   |                   |                   |                   |
> | Native            | 46.17             | 46.17             | 77.23             | 57.51             | 85.53             | 63.43             | 94.04             | 68.46             |
> | with GM Strategy  | **47.45 (+1.28)** | **47.45 (+1.28)** | **77.87 (+0.64)** | **58.16 (+0.65)** | **85.53 (+0.00)** | **63.81 (+0.38)** | 93.92 (-0.12)     | **68.68 (+0.22)** |
> | with PM Strategy  | **47.87 (+1.70)** | **47.87 (+1.70)** | **77.28 (+0.05)** | **58.26 (+0.75)** | **86.81 (+1.28)** | **64.71 (+1.28)** | **94.17 (+0.13)** | **69.49 (+1.03)** |
> | with EM Strategy  | **52.77 (+6.60)** | **52.77 (+6.60)** | **80.28 (+3.05)** | **62.02 (+4.51)** | **87.45 (+1.92)** | **66.98 (+3.55)** | 93.92 (-0.12)     | **71.30 (+2.84)** |
> | **LoCoMo**        |                   |                   |                   |                   |                   |                   |                   |                   |
> | Native            | 21.9              | 21.9              | 38.11             | 32.17             | 45.06             | 35.11             | 54.83             | 38.24             |
> | with GM Strategy  | **22.00 (+0.10)** | **22.00 (+0.10)** | **38.12 (+0.01)** | **32.22 (+0.05)** | **45.12 (+0.06)** | **35.15 (+0.04)** | **54.88 (+0.05)** | **38.30 (+0.06)** |
> | with PM Strategy  | **23.51 (+1.61)** | **23.51 (+1.61)** | **40.68 (+2.57)** | **34.33 (+2.16)** | **47.89 (+2.83)** | **37.22 (+2.11)** | **57.00 (+2.17)** | **40.39 (+2.15)** |
> | with EM Strategy  | **24.82 (+2.92)** | **24.82 (+2.92)** | **40.58 (+2.47)** | **34.57 (+2.40)** | **47.73 (+2.67)** | **37.64 (+2.53)** | **59.26 (+4.43)** | 41.37 (+3.13)     |
> | **LongMemEval-M** |                   |                   |                   |                   |                   |                   |                   |                   |
> | Native            | 35.53             | 35.53             | 62.17             | 44.47             | 70.21             | 49.14             | 82.98             | 54.48             |
> | with GM Strategy  | **36.81 (+1.28)** | **36.81 (+1.28)** | **62.55 (+0.38)** | **44.75 (+0.28)** | **71.06 (+0.85)** | **49.66 (+0.52)** | **83.19 (+0.21)** | **54.80 (+0.32)** |
> | with PM Strategy  | **37.02 (+1.49)** | **37.02 (+1.49)** | **64.17 (+2.30)** | **46.47 (+2.00)** | **73.40 (+3.19)** | **51.94 (+2.80)** | **84.26 (+1.28)** | **56.55 (+2.07)** |
> | with EM Strategy  | **43.40 (+7.87)** | **43.40 (+7.87)** | **69.15 (+6.98)** | **50.67 (+6.20)** | **76.17 (+5.96)** | **55.04 (+5.90)** | **85.96 (+2.98)** | **59.91 (+5.43)** |

---

> ### Author Response · Authors · 2025-11-21
> **Response to Reviewer ipaj [Part 2]**
>
> ## **2. Personal Semantic Memory Method**
>
> > **W2:** What justifies the keyword method adoption in personal semantic memory?
>
> Thanks for your question. **The keyword method is adopted to address the significant semantic gap between queries and memories in personal semantic memory, which is larger compared to general semantic memory**.  **The related statement has been added in `Section 2.3` on Page 5 in the revised version.**
>
> Personal queries often involve specific nuances, making precise retrieval difficult with fake memory generated by LLM. For example, the query "What is your occupation?",  there are many possible answers, e.g., teacher, doctor, student, painter, which may not be exactly matched through fake memories. To address this, we expand the query by generating related keywords using an LLM (e.g., expanding "occupation" to include " job" and "work") and appending them to the query. **This broadens the semantic scope, reducing the gap and improving retrieval accuracy**.
>
> ## **3. OOD Classification**
>
> > **W3:** I don't get a clear idea how the classifier will go for the OOD classification (does not fall into the classified types).
> >
> > **Q2:** How does the trained classifier handle and be trained for the classes that do not fall into existing categories?
>
> Thanks for your question, and we apologize for any confusion caused by the paper. For **NoneType memory, since queries only retrieve memory with types that appears in the fake memory, it will not occur in any query's retrieval scope**.  As shown in Table 2, NoneType memories typically lack meaningful information (e.g., generic questions or sentences). Filtering such memories prevents irrelevant retrievals, thereby improving retrieval efficiency.
>
> For queries without types, this scenario rarely occurs since the query's type is determined by the type appearing most frequently among the five fake memories generated by the LLM. **A query will only lack types if and only if all five generated fake memories are classified as NoneType**. In our experiments, this scenario has never been observed.
>
> **Table 2: The NoneType Classification Criterion.**
>
> | Memory Type |         Content Type         |     Example                        |
> | :---------: | :--------------------------: | :--------------------------------------------------: |
> |  NoneType   | Question without Information | Do you have any tips on how to organize a shoe rack? |
> |             | Sentence without information |          Hello; GoodBye, see you next time           |
>
> ## **4. Fake Memory Technology**
>
> > **Q1:** Is fake memory technology used for personal semantic memory as well?
>
> Thanks for the questions.  **Fake memories are indeed used to enhance the retrieval of personal semantic memories**. In fact, fake memories are employed in the retrieval of three types of memories, as they are generated during the query router phase. **Invoking fake memories requires no additional generation time.**
>
> ## **5. Classification Ablation Study**
>
> > **Q3:** In ablation study (classify), is the reported performance without classify but with key and fake strategies?
>
> Thanks for the questions. We apologize for any unclear expressions in the paper. In the ablation study for classification, **the reported performance did not utilize any retrieval enhancement strategies, and just filtered the memory corpus by query type.** **The related statement has been added in `Section 3.3` on Page 8 in the revised version.**
>
> As shown in Table 3 (corresponding to Figure 3 and Table 11 in the paper), the purpose of this ablation experiment is to **verify whether our type filtering strategy is effective**, i.e., **whether it successfully routes queries to the correct types**. Therefore, to eliminate interference from other strategies, no retrieval enhancement techniques were employed in this experiment.
>
> **Table 3: The ablation study of the proposed classification strategy on three datasets.**
>
> | Method  | Recall@1  | NDCG@1  | Recall@3  | NDCG@3    | Recall@5  | NDCG@5    | Recall@10 | NDCG@10   |
> | - | - | -- | - | - | - | - | - | - |
> | **LongMemEval-S** |     |   |    |   |   |    |   |   |
> | w/o Classify      | 43.40   | 43.40   | 73.62   | 54.00 | 83.83 | 60.33 | 94.26 | 65.79 |
> | w/ Classify       | **46.17** | **46.17** | **77.23** | **57.51** | **85.53** | **63.43** | **94.04** | **68.46** |
> | **LoCoMo**        |    |   |    |   |   |    |   |    |
> | w/o Classify      | 20.75     | 20.75     | 36.56 | 30.86 | 44.06 | 34.16| 53.93| 37.31 |
> | w/ Classify       | **21.90** | **21.90** | **38.11** | **32.17** | **45.06** | **35.11** | **54.83** | **38.24** |
> | **LongMemEval-M** |     |   |    |   |   |    |   |    |
> | w/o Classify      | 33.19 | 33.19  | 57.87 | 41.04| 68.09 | 46.46 | 82.98| 51.77|
> | w/ Classify       | **35.53** | **35.53** | **62.17** | **44.47** | **70.21** | **49.14** | **82.98** | **54.48** |

---

> ### Author Response · Authors · 2025-11-21
> **Response to Reviewer ipaj [Part 3]**
>
> ## **6. Hybrid Retrieval Ablation Study**
>
> > **Q4:** In ablation study (Hybrid), is keyword only impacting the personal semantic memory?
>
> Thanks for your feedback. We apologize for any confusion caused by the paper. **The “key” in Table 4 (corresponding to Table 4 in the paper) indicates that the keyword expansion strategy was applied to all memory**. The “fake memory” is similarly configured. The purpose of the Table 4 experiments was to validate **the advantages of hybrid retrieval strategies over single retrieval strategies**.  **The related statement has been added in `Section 3.3` on Page 8 in the revised version.**
>
> **Table 4: The ablation study of the proposed hybrid strategy on three datasets.**
>
> | Data         | **LongMemEval-S** |              |              | **LoCoMo**   |              |              | **LongMemEval-M** |              |              |
> | ------------ | ----------------- | ------------ | ------------ | ------------ | ------------ | ------------ | ----------------- | ------------ | ------------ |
> | **Metric**   | **Recall@1**      | **Recall@3** | **Recall@5** | **Recall@1** | **Recall@3** | **Recall@5** | **Recall@1**      | **Recall@3** | **Recall@5** |
> | Native       | 43.40             | 73.62        | 83.83        | 20.75        | 36.56        | 44.06        | 33.19             | 57.87        | 68.09        |
> | Key          | 43.82             | 75.10        | 84.25        | 20.90        | 37.06        | 44.61        | 33.19             | 60.63        | 66.59        |
> | Fake Memory  | 51.06             | 78.72        | 87.44        | 23.23        | 42.40        | 49.48        | 41.48             | 69.36        | 78.72        |
> | Our Strategy | **55.74**         | **81.06**    | **88.72**    | **26.54**    | **43.15**    | **50.60**    | **46.17**         | **71.91**    | **80.21**    |
>
> ## **7. Agent Memory Direction**
>
> > **Q5:** In related works agent memory, it seems that what the paper proposes can naturally enhance the existing agent memory techniques by letting agent possess/learn different strategies based on memory type. Do authors agree and have further thoughts to share on this?
>
> Thank you for your insightful comment. Yes, we completely agree with the reviewer’s perspective. In fact, the classification proposed in our paper is primarily inspired by **declarative memory** categorizations from cognitive psychology [1]. Declarative memory is a type of long-term memory involving conscious recollection, which includes **semantic memory (memory for facts)** and **episodic memory (memory for events)**. These often correspond to memories that can be "declared" or described, such as **dialogue memory** in conversation systems.
>
> Beyond declarative memory, cognitive psychology also defines another major category: **nondeclarative memory**, which aligns well with certain types of agent memory, such as logs of tool usage. Non-declarative memory includes **procedural memory**, **priming memory**, habituation, and so on.
>
> We believe that exploring valuable classifications of nondeclarative memory, alongside designing appropriate retrieval and generation strategies for each memory type, is a highly promising direction for advancing agent memory systems. This could not only enhance the agent’s ability to adapt and learn different strategies based on memory types but also offer a more cognitively aligned framework for memory design. Thank you again for raising this excellent point!
>
>  **The related statement has been added in `Section 4` on Page 10 in the revised version.**
>
> [1] Eysenck, Michael W., and Mark T. Keane. Cognitive psychology: A student's handbook. Psychology press, 2020
>
> > *We hope these responses address your concerns comprehensively. Feel free to let us know if you have any additional questions, and we are more than happy to provide further clarification on any aspect of our work.*
> >
> > *Respectfully and sincerely,*
> >
> > *The Authors*

---

> > ### Comment · Reviewer_ipaj · 2025-11-22
> >
> > Highly appreciate authors rebuttals that address all my comments with details and care.
> >
> > For question 1: I think the following is important for future readers to grasp "We also want to clarify a potential misunderstanding: the query type is determined by the type of the most frequent label among the five "fake memories" generated by the LLM. As such, the query itself is not multi-labeled. However, the retrieved memories may belong to multiple types or none at all."
> >
> > For OOD classification, since it is not relevant for the paper (i.e. not observed or used in the following RAG experimetns), I am wondering if there is a better way to write this to avoid confusion although I certainly admit that writing is hard.
> >
> > I do think that question 5 brings further grouding for the experimental results.
> >
> > Finally, for the agent memory direction, I was more thinking that today many industries and academics build agents, and they are all equipped with memory. The memory type that this paper proposes might be useful for these applications, I would encourage authors to think in this direction and make constructive proposals in the paper.

---

> > > ### Author Response · Authors · 2025-11-23
> > > **Response to Reviewer ipaj**
> > >
> > > **Dear Reviewer ipaj**:
> > >
> > > Thanks very much for your feedback! We highly appreciate your detailed comments, which have been invaluable in improving the clarity and quality of our work. Below, we address each of your points in detail:
> > >
> > > - **For Question 1:**
> > >
> > >   Thank you for pointing out the issue. **We have revised `Section 2.2` on Page 4 to include a clearer explanation of this concept**. We hope this additional explanation provides better clarity to readers.
> > >
> > > - **For the OOD classification issue:**
> > >
> > >   Thanks for the comments. To address this, we have provided further elaboration as follows:
> > >
> > >   As mentioned in the third point of the previous response, **"NoneType" memories typically correspond to questions with no information**, such as “What is your name?” or generic statements like “Hello,” “Good morning,” or “Nice to meet you.” These kinds of texts not only increase retrieval latency but may also interfere with correctly retrieving relevant content from the corpus. Therefore, **when constructing the TriMEM dataset, we intentionally included the NoneType type to filter out such meaningless inputs and improve retrieval performance**.
> > >
> > >   Regarding NoneType queries, as previously stated, the query type is conditional on the type that occurs most frequently among the generated fake memories. Since **queries in practical applications are often meaningful**, such as “What is John’s major?” and **fake memories should contain information to answer the given query, it is rare for LLMs to generate fake memories containing meaningless responses** like “Hello” or counter-questions like “What is your name?”, which is why NoneType queries did not occur in our experiments.
> > >
> > >   However, we believe that introducing additional modules that can process Nonetype query to ensure robustness is better. For instance, when a NoneType query is detected, we could default to using basic dense retrieval without further processing. **We have added the related statement in `Appendix D.4` on Page 23 to clarify the OOD Classification issue**.
> > >
> > >   We hope these statements will reduce your confusion. Once again, we appreciate your valuable feedback on this matter.
> > >
> > > - **For the agent memory direction:**
> > >
> > >   Thank you for suggesting this direction! Currently, **agent memory paradigms can generally be divided into two categories**:
> > >
> > >   - **Retrieval-augmented generation (RAG)-based methods**, such as A-Mem [1], Mem0 [2],  and MIRIX [3].
> > >
> > >   - **Reinforcement learning (RL)-based methods**, such as Memory-R1 [4] and Mem-$\alpha$ [5].
> > >
> > >   **For the RAG-based paradigm, our method can be seamlessly integrated since it is fundamentally a RAG pipeline**. Our approach simply pre-trains a router model to determine the type of memory and query, which can easily be incorporated into existing frameworks.
> > >
> > >   **For the RL-based paradigm, we believe our router model could also be incorporated without retraining the entire system**. Specifically, the router model could be used within the RL training loop, with retrieval performance metrics set as the reward signal. This would allow the model to adaptively learn the categories of memory and query, as well as select the most appropriate retrieval strategy.
> > >
> > >   Besides, as mentioned in Point 7 of the previous response, our work primarily focuses on dialogue memory types. If agent memory systems need to retrieve tool-invoked logs or working memories, it would be necessary to introduce additional categories, such as non-declarative memories, to ensure broader applicability.
> > >
> > > **Thanks again for your feedback. We hope these statements can resolve your confusion.  Feel free to let us know if you have any additional questions, and we are very happy to provide further clarification on any aspect of our work.**
> > >
> > > Respectfully and sincerely,
> > >
> > > The Authors
> > >
> > > [1] Xu, Wujiang, et al. "A-mem: Agentic memory for llm agents." *arXiv preprint arXiv:2502.12110* (2025).
> > >
> > > [2] Chhikara, Prateek, et al. "Mem0: Building production-ready ai agents with scalable long-term memory." *arXiv preprint arXiv:2504.19413* (2025).
> > >
> > > [3] Wang, Yu, and Xi Chen. "Mirix: Multi-agent memory system for llm-based agents." *arXiv preprint arXiv:2507.07957* (2025).
> > >
> > > [4] Yan, Sikuan, et al. "Memory-r1: Enhancing large language model agents to manage and utilize memories via reinforcement learning." *arXiv preprint arXiv:2508.19828* (2025).
> > >
> > > [5] Wang, Yu, et al. "Mem-{\alpha}: Learning Memory Construction via Reinforcement Learning." *arXiv preprint arXiv:2509.25911* (2025).

---

### Official Review · Reviewer_gzEA · 2025-10-23

**Soundness:** 2
**Presentation:** 3
**Contribution:** 2
**Rating:** 2
**Confidence:** 4

**Summary:**

The paper targets the memory retrieval and response generation problem, and propose a memory type classficiation benchmark by classifing conversational history of existing benchmarks into different types of memory i.e., episodic memory and personal semantic memory, and advocate to retrieve related memory according to the required type of memory (by classifing the type of query using the pre-trained model on the ccollected benchmark), and design specific retrieval strategies and pruning strategy to improve the retrieval and generation quality. Generally, the benchmark is build on top of existing benchmarks, and proposed method is more like re-combination of existing techniques, and more focus on retrieval part. Despite the experimental results is promising, it is uncertain where these gains come from.

**Strengths:**

1. The proposed method is reasonable desipte it is too specfic for each type and more like engineering tricks.
2. The experimental results confirm the effectiveness of proposed method.

**Weaknesses:**

1. There are many studies that focus on different types of memory and propose to retrieve according to determined type [1]. This significantly weaken the contribution and novelty of the proposed method.

2. the constructed benchmark is not detailed, i.e., how do you do quality control, why choose these benchmarks? these details are not mentioned in main paper. Despite some are included in appendix, it is not comprehensive.

2. Generally, the proposed method contains three parts: query ruoter (a.k.a, part a, to decide which type of memory to retrieve), memory retrieval according to label by part a (a.k.a., part b), and memory pruning (a.k.a., part c). the experiment only confirms the effectiveness of a+b+c, and each module under this system. It is not clear: i) whether any method incorporate independant part leads to better performance, i.e., query router + other retrieval strategies in the baselines; ii) the effectes of cascade errors; iii) the results of table 4 and 5 shows there is no significant gain for part b and part c.


[1] Perltqa: A personal long-term memory dataset for memory classification, retrieval, and fusion in question answering.

**Questions:**

see weakness.

---

> ### Author Response · Authors · 2025-11-21
> **Response to Reviewer gzEA [Part 1-1]**
>
> **Dear Reviewer gzEA**:
>
> Thank you so much for your thoughtful review and valuable feedback. We sincerely appreciate the time and effort you devoted to evaluating our work. Below, we have provided detailed responses to your comments.
>
> ## **1. Novelty of the Method**
>
> > **W1:** There are many studies that focus on different types of memory and propose to retrieve according to determined type [1]. This significantly weaken the contribution and novelty of the proposed method. **The statement has been added in `Section 2.2` on Page 3 in the revised version.**
>
> Thanks for your valuable feedback. However, **we believe there exists misunderstandings regarding both our work MemoType and the referenced work PerLTQA [1]**. Below, we provide clarifications from both the **Methodological** and **Data** perspectives.
>
> ### **1.1 Methodological Perspective**
>
> We appreciate your concerns about the novelty of our method. However, **as clarified in the** **[Memory Retrieval subsection](https://arxiv.org/pdf/2402.16288)** **on page 5**, **PerLTQA simply retrieves the top k memories for each memory category without proposing any category filtering or category-specific retrieval enhancement strategies**. In contrast, MemoType introduces several innovative mechanisms in methodological perspective, as detailed below:
>
> - **Memory and Query Routing:** MemoType finetunes a router model guided by the TriMEM dataset to adaptively route memory. For query routing, MemoType utilizes the extensive knowledge of large language models (LLMs) to generate multiple "fake memories" for each query and employs a voting mechanism to improve query classification accuracy. **By retrieving memories aligned with the query's type instead of the entire memory corpus, MemoType enhances retrieval efficiency.**
> - **Type-specific Retrieval Mechanisms:** MemoType designs tailored retrieval strategies for different memory types. For instance, Episodic memory: We decouple time, person, and location elements using an event-matching method. Personal semantic memory: We employ synonym expansion to bridge semantic gaps. General semantic memory: We generate supplementary "fake memories" based on LLM-embedded knowledge.
>
> In addition, **we have examined other works related to memory type and highlighted the fundamental differences between them and MemoType**. For instance, MIRIX [2] introduces six memory categories and employs large language models (LLMs) to classify memories. However, similar to the method used in PerLTQA, MIRIX retrieves the top k memories for each type without proposing any retrieval enhancement strategies tailored to memory types. Similarly, MMS [3] extracts four types of memories from a corpus using LLMs but relies solely on standard dense retrieval methods.
>
> **To the best of our knowledge, MemoType is the first memory framework that adaptively classifies dialogue memory and designs distinct retrieval and storage mechanisms for different types of memory**.
>
> **If the reviewer knows any other works closely related to ours, we warmly invite further discussion.**
>
> [1] Du, Yiming, et al. "PerLTQA: A personal long-term memory dataset for memory classification, retrieval, and fusion in question answering." *Proceedings of the 10th SIGHAN Workshop on Chinese Language Processing (SIGHAN-10)*. 2024.
>
> [2] Wang Y, Chen X. Mirix: Multi-agent memory system for llm-based agents[J]. arXiv preprint arXiv:2507.07957, 2025.
>
> [3]  Zhang, Gaoke, et al. "Multiple memory systems for enhancing the long-term memory of agent." arXiv preprint arXiv:2508.15294 (2025).

---

> ### Author Response · Authors · 2025-11-21
> **Response to Reviewer gzEA [Part 1-2]**
>
> ### **1.2 Data Perspective**
>
> Although PerLTQA is a pioneering benchmark research in the memory classification field, it still has several limitations which are given preliminary solutions in the TriMEM we constructed.
>
> - **Single Memory Type.**  PerLTQA first determines the memory category before generating the corresponding memory, which leads to a problem: **the generated memory only has a single label**. In real-world scenarios, a single conversation may involve multiple topics, resulting in multiple memory categories. Single-label memories fail to reflect real-world scenarios and artificially reduce classification difficulty. In the proposed TriMEM dataset, we constructed memories from real scenarios first, then annotated and refined three different memory types labels, which avoids the single memory type issue.
>
> - **Incomplete Memory Type.** PerLTQA [1] predefines four information types (personal profiles, social relationships, events, dialogues). Among these, personal profiles and social relationships are categorized as semantic memory, while events and dialogues are categorized as episodic memory. However, such classification doesn't cover all memories:
>
>   **i. Incomplete Semantic Memory.** The semantic memory in PerLTQA consists solely of personal semantic memory, **neglecting general semantic memory**. Referring to the cognitive psychology [4] definition of semantic memory: "A form of long-term memory consisting of general knowledge about the world, concepts, language, and so on" (Definition on Page 285 in [4]), general world knowledge and other memories unrelated to the user also belong to the semantic memory domain.
>
>   **ii. Incomplete Episodic Memory.**The episodic memory in PerLTQA focuses solely on past specific events, representing retrospective episodic memory, while **lacking prospective episodic memory** for future planning (Definition on Page 367 in [4] ).
>
>   In the constructed TriMEM dataset, referring to the memory classification principle in cognitive psychology [4], we comprehensively consider different memory categories. More detailed classification can be found in Table 1 in our paper.
>
> - **Coarse-Grained Memory Type.** **PerLTQA uniformly classifies all dialogues as episodic memory**. Such coarse-grained categorization is unreasonable. Dialogues are crucial retrieval objects in existing memory scenarios. For instance, in the two common memory benchmarks LoCoMo [5] and LongMemEval [6], **dialogues are the primary and exclusive retrieval objects**. Moreover, in real-world scenarios, accessible data typically exists in dialogue form, while user profiles and social relationships are often difficult to obtain. Therefore, **the distinction between user profiles and dialogues as separate types lacks transferability in practical memory scenarios**.
>
> **To the best of our knowledge, TriMEM is the first dialogue memory multi-classification dataset.**
>
> [1] Du, Yiming, et al. "PerLTQA: A personal long-term memory dataset for memory classification, retrieval, and fusion in question answering." *Proceedings of the 10th SIGHAN Workshop on Chinese Language Processing (SIGHAN-10)*. 2024.
>
> [4] Eysenck, Michael W., and Mark T. Keane. Cognitive psychology: A student's handbook. Psychology press, 2020.
>
> [5] Maharana A, Lee D H, Tulyakov S, et al. Evaluating Very Long-Term Conversational Memory of LLM Agents[C]//Proceedings of the 62nd Annual Meeting of the Association for Computational Linguistics (Volume 1: Long Papers). 2024: 13851-13870.
>
> [6] Wu D, Wang H, Yu W, et al. LongMemEval: Benchmarking Chat Assistants on Long-Term Interactive Memory[C]//The Thirteenth International Conference on Learning Representations.

---

> ### Author Response · Authors · 2025-11-21
> **Response to Reviewer gzEA [Part 2]**
>
> ## **2. Supplementary Explanations for Data Construction**
>
> > **W2:** the constructed benchmark is not detailed, i.e., how do you do quality control, why choose these benchmarks? these details are not mentioned in main paper. Despite some are included in appendix, it is not comprehensive.
>
> Thanks for your valuable feedback. We apologize for the lack of clarity in the original submission. Below, we address your specific concerns regarding quality control and the rationale behind the choice of benchmarks. **The statement has been added in `Appendix A.3` on Page 15-16 in the revised version.**
>
> - **Quality Control**
>
>   To ensure high labeling accuracy and consistency, we employed a two-step process. First, we used the GPT-4o-2024-11-20 model to annotate all categories of data based on three carefully designed prompts. These prompts were meticulously crafted and validated on a subset of the data to ensure their effectiveness for the classification task. Notably, we opted for separate prompts for each category, as this approach achieved higher classification accuracy compared to using a single prompt for all categories. Following the automated labeling process, each category’s labels underwent manual validation by a dedicated reviewer to rectify any errors and ensure the correctness of the final labels. This combination of automated and manual processes guaranteed the reliability and accuracy of the annotations.
>
> - **Benchmark Selection**
>
>   As mentioned in the "Data Perspective" section of our paper, we aimed to annotate memory categories in real-world dialogue scenarios rather than generating memory-related dialogues from predefined categories, which would risk introducing bias by over-focusing on single categories. We chose **LoCoMo**  [5] and **LongMemEval** [6] as they are widely recognized memory datasets frequently referenced in works such as A-Mem [7], mem0 [8], MIRIX [2], and Secom [9]. This highlights their significance and credibility. LongMemEval emphasizes user-assistant interactions, while LoCoMo focuses on user-user dialogues, aligning with our goal of covering diverse memory scenarios and topics. To further enhance the generalizability of our annotated dataset beyond evaluation-specific categories, we included **PersonaMem** [10], a personalized memory benchmark that introduces varied user profiles. This addition strengthens the dataset’s applicability across broader dialogue classification tasks.
>
> We hope this provides a clearer understanding of our methodology and decisions. Please let us know if further clarifications are needed.
>
> [2] Wang Y, Chen X. Mirix: Multi-agent memory system for llm-based agents[J]. arXiv preprint arXiv:2507.07957, 2025.
>
> [5] Maharana A, Lee D H, Tulyakov S, et al. Evaluating Very Long-Term Conversational Memory of LLM Agents[C]//Proceedings of the 62nd Annual Meeting of the Association for Computational Linguistics (Volume 1: Long Papers). 2024: 13851-13870.
>
> [6] Wu D, Wang H, Yu W, et al. LongMemEval: Benchmarking Chat Assistants on Long-Term Interactive Memory[C]//The Thirteenth International Conference on Learning Representations.
>
> [7] Xu, Wujiang, et al. "A-mem: Agentic memory for llm agents." *arXiv preprint arXiv:2502.12110* (2025).
>
> [8] Chhikara, Prateek, et al. "Mem0: Building production-ready ai agents with scalable long-term memory." *arXiv preprint arXiv:2504.19413* (2025).
>
> [9] Pan, Zhuoshi, et al. "Secom: On memory construction and retrieval for personalized conversational agents." *The Thirteenth International Conference on Learning Representations*. 2025.
>
> [10] Jiang, Bowen, et al. "Know me, respond to me: Benchmarking llms for dynamic user profiling and personalized responses at scale." *arXiv preprint arXiv:2504.14225* (2025).

---

> ### Author Response · Authors · 2025-11-21
> **Response to Reviewer gzEA [Part 3-1]**
>
> ## **3. Ablation Study of the Proposed Module**
>
> > **W3:** Generally, the proposed method contains three parts: query ruoter (a.k.a, part a, to decide which type of memory to retrieve), memory retrieval according to label by part a (a.k.a., part b), and memory pruning (a.k.a., part c). the experiment only confirms the effectiveness of a+b+c, and each module under this system. It is not clear: i) whether any method incorporate independant part leads to better performance, i.e., query router + other retrieval strategies in the baselines; ii) the effectes of cascade errors; iii) the results of table 4 and 5 shows there is no significant gain for part b and part c
>
> Thanks for your insightful comments. We have supplemented additional experiments below. **The experiment and related statement have been added in `Table 10` on Page 24 and  `Appendix D.1` on Page 22 in the revised version.**
>
> - As shown in **Table 1**, we evaluated **the query router (Part A) combined with various retrieval strategies from baselines**. **The results indicate that most combinations lead to performance improvements**. Among these, the EM (episodic memory) strategy achieves the most significant gains, highlighting the efficacy of combining the query router with advanced retrieval strategies.
>
> **Table 1: The ablation study of different retrieval strategies on three datasets.**
>
> | Method                     | Recall@1          | NDCG@1            | Recall@3          | NDCG@3            | Recall@5          | NDCG@5            | Recall@10         | NDCG@10           |
> | -------------------------- | ----------------- | ----------------- | ----------------- | ----------------- | ----------------- | ----------------- | ----------------- | ----------------- |
> | **LongMemEval-S**          |                   |                   |                   |                   |                   |                   |                   |                   |
> | Query Route                | 46.17             | 46.17             | 77.23             | 57.51             | 85.53             | 63.43             | 94.04             | 68.46             |
> | Query Route + GM Strategy  | **47.45 (+1.28)** | **47.45 (+1.28)** | **77.87 (+0.64)** | **58.16 (+0.65)** | **85.53 (+0.00)** | **63.81 (+0.38)** | 93.92 (-0.12)     | **68.68 (+0.22)** |
> | Query Route +  PM Strategy | **47.87 (+1.70)** | **47.87 (+1.70)** | **77.28 (+0.05)** | **58.26 (+0.75)** | **86.81 (+1.28)** | **64.71 (+1.28)** | **94.17 (+0.13)** | **69.49 (+1.03)** |
> | Query Route +  EM Strategy | **52.77 (+6.60)** | **52.77 (+6.60)** | **80.28 (+3.05)** | **62.02 (+4.51)** | **87.45 (+1.92)** | **66.98 (+3.55)** | 93.92 (-0.12)     | **71.30 (+2.84)** |
> | **LoCoMo**                 |                   |                   |                   |                   |                   |                   |                   |                   |
> | Query Route                | 21.9              | 21.9              | 38.11             | 32.17             | 45.06             | 35.11             | 54.83             | 38.24             |
> | Query Route +  GM Strategy | **22.00 (+0.10)** | **22.00 (+0.10)** | **38.12 (+0.01)** | **32.22 (+0.05)** | **45.12 (+0.06)** | **35.15 (+0.04)** | **54.88 (+0.05)** | **38.30 (+0.06)** |
> | Query Route +  PM Strategy | **23.51 (+1.61)** | **23.51 (+1.61)** | **40.68 (+2.57)** | **34.33 (+2.16)** | **47.89 (+2.83)** | **37.22 (+2.11)** | **57.00 (+2.17)** | **40.39 (+2.15)** |
> | Query Route +  EM Strategy | **24.82 (+2.92)** | **24.82 (+2.92)** | **40.58 (+2.47)** | **34.57 (+2.40)** | **47.73 (+2.67)** | **37.64 (+2.53)** | **59.26 (+4.43)** | **41.37 (+3.13)** |
> | **LongMemEval-M**          |                   |                   |                   |                   |                   |                   |                   |                   |
> | Query Route                | 35.53             | 35.53             | 62.17             | 44.47             | 70.21             | 49.14             | 82.98             | 54.48             |
> | Query Route +  GM Strategy | **36.81 (+1.28)** | **36.81 (+1.28)** | **62.55 (+0.38)** | **44.75 (+0.28)** | **71.06 (+0.85)** | **49.66 (+0.52)** | **83.19 (+0.21)** | **54.80 (+0.32)** |
> | Query Route +  PM Strategy | **37.02 (+1.49)** | **37.02 (+1.49)** | **64.17 (+2.30)** | **46.47 (+2.00)** | **73.40 (+3.19)** | **51.94 (+2.80)** | **84.26 (+1.28)** | **56.55 (+2.07)** |
> | Query Route +  EM Strategy | **43.40 (+7.87)** | **43.40 (+7.87)** | **69.15 (+6.98)** | **50.67 (+6.20)** | **76.17 (+5.96)** | **55.04 (+5.90)** | **85.96 (+2.98)** |      **59.91 (+5.43)**             |

---

> ### Author Response · Authors · 2025-11-21
> **Response to Reviewer gzEA [Part 3-2]**
>
> - **We disagree with the notion that performance improvements primarily stem from the query router**. As illustrated in **Tables 2 and 3**, hybrid strategies contribute significantly to performance, especially for challenging metrics like Recall@1. For instance, in LongMemEval-S, the hybrid approach improves Recall@1 by **12.34%** compared to the native strategy, with an average gain of **8.22%** across all metrics. **Even compared to the suboptimal fake memory strategy, the hybrid strategy achieves higher gains than the query router alone**. These results emphasize the importance of Part B in boosting overall retrieval effectiveness.
>
> **Table 2: The Retrieval Improvement of the proposed classification strategy on three datasets. "Imp" means "Improvement".**
>
> | Strategy| LongMemEvalS |  |  |  | LoCoMo |  |  |  | LongMemEvalM |  |  |  |
> |:---:|---|---|:---:|---|---|---|:---:|---|:---:|---|---|---|
> | Metric | Recall@1 | Recall@3 | Recall@5 | Average | Recall@1 | Recall@3 | Recall@5 | Average | Recall@1 | Recall@3 | Recall@5 | Average |
> | w/o Classify | 43.40 | 73.62 | 83.83 | 66.95 | 20.75 | 36.56 | 44.06 | 33.79 | 33.19 | 57.87 | 68.09 | 53.05 |
> | w/ Classify | 46.17 | 77.23 | 85.53 | 69.64 | 21.90 | 38.11 | 45.06 | 35.02 | 35.53 | 62.17 | 70.21 | 55.97 |
> | **Imp** |     2.77      | 3.61     | 1.70     | **2.69** | 1.15     | 1.55     | 1.00     | **1.23** | 2.34          |   4.30   | 2.12     | **2.92** |
>
> **Table 3: The Retrieval Improvement of the proposed hybrid strategy on three datasets. "Imp" means "Improvement"**
>
> | Strategy | LongMemEvalS |  |  |  | LoCoMo |  |  |  | LongMemEvalM |  |  |  |
> |:---:|---|---|---|:---:|---|---|---|:---:|---|---|---|:---:|
> | Metric | Recall@1 | Recall@3 | Recall@5 | Average | Recall@1 | Recall@3 | Recall@5 | Average | Recall@1 | Recall@3 | Recall@5 | Average |
> | Native | 43.40 | 73.62 | 83.83 | 66.95 | 20.75 | 36.56 | 44.06 | 33.79 | 33.19 | 57.87 | 68.09 | 53.05 |
> | Key | 43.82 | 75.10 | 84.25 | 67.72 | 20.90 | 37.06 | 44.61 | 34.19 | 33.19 | 60.63 | 66.59 | 53.47 |
> | Fake Memory | 51.06 | 78.72 | 87.44 | 72.41 | 23.23 | 42.40 | 49.48 | 38.37 | 41.48 | 69.36 | 78.72 | 63.19 |
> | Our Strategy | 55.74 | 81.06 | 88.72 | 75.17 | 26.54 | 43.15 | 50.60 | 40.10 | 46.17 | 71.91 | 80.21 | 66.10 |
> |   **Imp over Native**    | 12.34 | 7.44     | 4.89     | **8.22** | 5.79     | 6.59     | 6.54     | **6.31** | 12.98         | 14.04    | 12.12    | **13.05** |
> | **Imp over Suboptimal** | 4.68| 2.34     | 1.28     | **2.77** | 3.31     | 0.75     | 1.12     | **1.73** | 4.69          | 2.55     | 1.49 | **2.91**  |
> - **The original version of the paper only displayed partial generation metrics, failing to illustrate the effectiveness of the pruning strategy.**  To address this, we supplemented the generation evaluation metrics, as shown in Table 4. Incorporating pruning (Part C) leads to significant improvements in generative metrics. For instance, in LongMemEval-S, pruning improves F1 (from 17.43 to 20.27) and BLEU (from 3.15 to 4.52). **While the GPT4J model in LongMemEval-M shows a minor decline, all other metrics exhibit notable improvements, underlining the value of pruning for generative tasks**
>
> **Table 4: The Generation Improvement of the proposed running strategy on three datasets. "Imp" means "Improvement"**
> | **Strategy** | **LongMemEval-S** |  |  |  |  |  |  |  |
> |:---:|:---:|:---:|:---:|:---:|:---:|:---:|:---:|:---:|
> | **Metric** | **GPT4J** | **F1** | **BLEU** | **Rouge1** | **Rouge2** | **RougeL** | **RougeLsum** | **BERTScore** |
> | w/o Prun | 49.00 | 17.43 | 3.15 | 18.15 | 8.87 | 16.62 | 16.83 | 82.88 |
> | w/ Prun | **50.00** | **20.27** | **4.52** | **21.13** | **10.49** | **19.54** | **19.71** | **85.32** |
> |  | **LoCoMo** |  |  |  |  |  |  |  |
> | w/o Prun | 39.88 | 18.45 | 4.18 | 18.92 | 9.58 | 17.81 | 17.82 | 85.19 |
> | w/ Prun | **40.38** | **19.69** | **4.53** | **20.10** | **10.08** | **18.91** | **18.94** | **85.43** |
> |  | **LongMemEval-M** |  |  |  |  |  |  |  |
> | w/o Prun | **42.00** | 15.99 | 2.80 | 16.81 | 8.10 | 15.18 | 15.44 | 84.44 |
> | w/ Prun | 41.60 | **17.79** | **3.78** | **18.73** | **8.72** | **17.14** | **17.29** | **85.02** |
>
> > *We hope these responses address your concerns comprehensively. Feel free to let us know if you have any additional questions, and we are more than happy to provide further clarification on any aspect of our work.*
> >
> > *Respectfully and sincerely,*
> >
> > *The Authors*

---

> > ### Comment · Reviewer_gzEA · 2025-11-25
> >
> > Thank you for detailed response. I decide to raise my initial score to 4. However, the novelty and contribution of this paper still is weaken by related work PerLTQA, I strongly disagree that `PerLTQA simply retrieves the top k memories for each memory category without proposing any category filtering or category-specific retrieval enhancement strategies`. Please read that paper carefully, it is clear in the task definition and table 3 that category filtering or category-specific retrieval enhancement strategies are included.

---

> ### Author Response · Authors · 2025-11-26
>
> **Dear Reviewer gzEA**,
>
> Thank you very much for your **score improvement** and for providing us with the opportunity to discuss! We apologize for any misunderstanding caused by the previous statement. We would like to continue clarifying some statements regarding PerLTQA [1]. In particular, we wish to further explain **the retrieval strategy used in PerLTQA,  which includes three main strategies**:
>
> 1. **Retrieving k memories from two types**, resulting in a total of 2k memories, as shown in **Eq.(2)**. `No category-aware strategies are applied`, and the retrieval is fixed to retrieve a fixed number of memories per category.
> 2. **Weighted scoring based on classification confidence** is applied to the 2k memories, as shown in **Eq.(3)**. `No category-specific retrieval enhancement strategies are used for semantic or episodic memories`, and the focus is merely on improving the relative ranking of more confidently classified memories.
> 3. **Selecting top-k memories from the retrieved 2k memories** for memory filtering. However, we believe that `this filtering mechanism has some significant issues`.
>
>    - The first term in Eq. (3) is based on the **classification confidence**. **This score is a fixed value, independent of the query and only dependent on whether the memory has certain category features**. Memories with stronger semantic and episodic features will always score higher, but this does not necessarily reflect whether they are the correct answer to the given query.
>
>    - **This issue becomes more pronounced when the dataset is harder to retrieve**. For example, in the LoCoMo dataset, which is challenging for retrieval, most methods achieve a recall@5 below 50%, indicating that **the second term in Eq. (3) lacks sufficient discriminative power to distinguish between the correct and incorrect memories**. In such cases, **the score in Eq. (3) is dominated by the first term, the classification confidence**. `For each query, the top-k retrieved memories will always be those with obvious category features, which could cause catastrophic performance degradation and is fundamentally different from our filtering strategy based on the query type`.
>
> We supplemented `the original text` in **the Memory Retrieval subsection of the Task Definition on page 5 of [PerLTQA](https://arxiv.org/pdf/2402.16288)**：
>
> ----------
> > Section 3.4 Task Definition
>
> ### **Memory Retrieval**
>
> We aim to perform memory retrieval by extracting relevant character memories for a given evaluation question from the PerLT memory database $M$, formalized as Eq.(2).
>
> $ m, s = R(q, M, k)$       $\(\text{2}\) $
>
> where $m$ is the retrieved memory with size $k$, and $s$ is the corresponding scores, $R$ is the retrieval model.
>
> **Our method distinguishes itself by initially retrieving $k$ memories from each category within the memory database**, amassing $2k$ potential memory candidates. These candidates undergo a re-ranking process influenced by their classification scores, culminating in a composite score for each memory $m_i$, which is computed as follows:
>
> $s'_i = \alpha \cdot P(\pi | m_i) + \beta \cdot \text{sigmoid}(s_i) $         $\(\text{3}\) $
>
> where **$P(\pi | m_i)$ is the probability given by the classification model that the memory item $m_i$ belongs to $\pi$**. The top $k$ memories are then selected based on these final scores. $\alpha$ and $\beta$ represent the weight of each term, and we set both to 0.5 to balance their contributions.
>
> --------------
>
> **If we have any errors or omissions in our statements about PerLTQA, we welcome the reviewer to point them out. We are keen to have the opportunity to discuss this with you again.**
>
> **Thanks again for your feedback and for providing us with an opportunity to clarify these points. We sincerely hope the above clarifications resolve your concerns.**
>
> Respectfully and sincerely,
>
> The Authors
>
> > [1] Du, Yiming, et al. "PerLTQA: A personal long-term memory dataset for memory classification, retrieval, and fusion in question answering." Proceedings of the 10th SIGHAN Workshop on Chinese Language Processing (SIGHAN-10). 2024.

---

### Official Review · Reviewer_Ajqg · 2025-10-31

**Soundness:** 3
**Presentation:** 3
**Contribution:** 3
**Rating:** 4
**Confidence:** 3

**Summary:**

This
 paper investigates how Large Language Models (LLMs) handle long-term
memory, particularly in Retrieval-Augmented Generation (RAG) systems
that store and recall past information to mitigate the “forgetting”
problem. Existing RAG-based memory frameworks typically apply the same
retrieval and storage strategy to all memories, overlooking the fact
that different types of memories (e.g., factual, episodic, or semantic)
require different handling. To address this, the authors introduce
TriMEM, a benchmark containing 6,000 annotated dialogue samples that
categorize diverse memory types across topics and scenarios. Building on
 this benchmark, they propose MemoType, an adaptive memory framework
that automatically identifies the type of each memory and applies
tailored retrieval and storage strategies accordingly. Experiments on
retrieval and generation tasks demonstrate that MemoType improves both
memory organization and overall model performance, highlighting the
importance of memory categorization in long-term language model
reasoning.

**Strengths:**

**Novel conceptual framing:**
  The paper introduces a meaningful and intuitive perspective by
categorizing memories in LLMs into distinct types (episodic, semantic,
factual) and applying **type-specific retrieval and storage
strategies**, addressing an underexplored dimension of long-term memory
modeling in LLMs.

* **New benchmark contribution (TriMEM):**
  The authors contribute a well-structured and valuable dataset,
**TriMEM**, containing 6,000 annotated dialogue samples with explicit
memory-type labels. This benchmark fills an important gap for studying
memory categorization and adaptive retrieval mechanisms in
conversational settings.

* **Clear motivation and design:**
  The problem formulation is well-motivated, and the overall pipeline —
from memory categorization to retrieval and generation — is logically
presented and easy to follow.

* **Potential for generalization and integration:**
  The proposed **MemoType** framework is modular and could, in
principle, be integrated into broader **RAG** or **agent-memory**
systems, making it a promising direction for long-term dialogue
reasoning and adaptive retrieval research.

**Weaknesses:**

**Limited validation of core claim:**
  The paper’s main contribution lies in classifying memory types
(episodic, semantic, factual) and applying type-specific retrieval
strategies. However, the benchmarks used do not contain explicit labels
for memory types. As a result, the experiments only demonstrate that
MemoType improves downstream performance, without directly showing that
the type-adaptive retrieval mechanism itself is responsible for these
gains.

* **Insufficient diversity of evaluation benchmarks:**
  The evaluation on LongMemEval-S, LongMemEval-M, and LoCoMo effectively
 measures retrieval and generation under long-context and multi-session
conditions. However, to substantiate claims of general effectiveness in
memory categorization and adaptive retrieval, the study would benefit
from incorporating **personalized or task-oriented memory benchmarks**
(e.g., PerLTQA, MEMTRACK). These settings better reflect realistic agent
 memory use cases—such as user profiles, preferences, or tool-use
history—where the distinction between factual, episodic, and procedural
memory is most impactful.

* **Questionable generalization of the memory-type classifier:**
  A notable concern lies in the **credibility and generalization** of
the proposed memory-type classifier, especially given that the results
in Table 9 show it outperforming significantly larger models such as
**Qwen3-8B, Qwen3-32B, Gemini-2.5-Flash, and GPT-4o-Mini**. The
classifier is a simple **BERT-based model** trained with binary
cross-entropy loss over only a few iterations, which raises doubts about
 how such a lightweight model achieves superior results. This suggests
the evaluation setup may be benchmark-specific rather than reflecting
true generalization. Without cross-domain tests, ablations, or analysis
of potential data leakage, the claims about robustness and real-world
applicability remain unconvincing.

**Questions:**

1. How were the memory-type labels (episodic, semantic, factual) assigned
or validated during training, given that existing benchmarks do not
provide such annotations?

2. Can the authors provide evidence or analysis showing that the
observed performance improvement specifically arises from type-adaptive
retrieval, rather than general architectural or training advantages?

3. Why were personalized or task-oriented benchmarks (e.g., PerLTQA,
MEMTRACK) not included in evaluation, given their relevance to memory
categorization and realistic long-term interaction scenarios?

4. How does the simple BERT-based classifier generalize beyond the TriMEM dataset? Have the authors tested it on out-of-domain data or with alternative encoders (e.g., DeBERTa, RoBERTa) to assess robustness?

5. The paper reports outperforming much larger models (e.g., GPT-4o-Mini, Qwen3-32B). Could the authors clarify the evaluation
protocol and whether there are benchmark-specific advantages or data  overlaps that might explain this result?

---

> ### Author Response · Authors · 2025-11-21
> **Response to Reviewer Ajqg [Part 1]**
>
> **Dear Reviewer Ajqg**:
>
> Thank you so much for your thoughtful review and valuable feedback. We sincerely appreciate the time and effort you devoted to evaluating our work. Below, we have provided detailed responses to your comments.
>
> ## **1. Label Assignment and Validation**
>
> > **Q1**: How were the memory-type labels (episodic, semantic, factual) assigned or validated during training, given that existing benchmarks do not provide such annotations?
>
> Thanks for the insightful comment, and we apologize for any confusion caused by the paper. Below, we will answer your questions from two aspects: **Type Assignment** and **Type Validation**:
>
> ### **1.1 Type Assignment**
>
> - **Memory Type Router**:
>   We employed a router model to classify memories into the three types. This router model was **pre-trained on the well-annotated TriMEM** dataset, achieving desirable memory classification performance. (**Note: The router model does not require additional training on the LongMemEval and LoCoMo datasets**) By doing so, MemoType adaptively determined the memory type for each memory.
> - **Query Type Router**:
>   For each query, we utilized a Large Language Model (LLM) to generate five fake memory candidates. The query type was determined based on the memory type that occurred most frequently among these candidates through a majority voting mechanism. Types appearing in false memories will be identified as the retrieval scope for the query. **This adaptive process effectively confined the retrieval scope to the most relevant memory type, thereby improving retrieval efficiency and accuracy**.
>
> ### **1.2 Type Validation**
>
> Since the dataset lacks true labels for memory types, we primarily validate the effectiveness of the classification strategy through ablation experiments. Specifically, we design two baseline settings: **"w/Classify" indicates filtering the memory corpus using our query labels; "w/o Classify" indicates retrieving all memories without category filtering**, as in the experiments shown in Figure 3 on page 7 and Table 11 on page 26 of our paper. The results are as follows:
>
> **Table 1: The ablation study of the proposed classification strategy on three datasets.**
>
> | Method            | Recall@1  | NDCG@1    | Recall@3  | NDCG@3    | Recall@5  | NDCG@5    | Recall@10 | NDCG@10   |
> | ----------------- | --------- | --------- | --------- | --------- | --------- | --------- | --------- | --------- |
> | **LongMemEval-S** |           |           |           |           |           |           |           |           |
> | w/o Classify      | 43.40     | 43.40     | 73.62     | 54.00     | 83.83     | 60.33     | 94.26     | 65.79     |
> | w/ Classify       | **46.17** | **46.17** | **77.23** | **57.51** | **85.53** | **63.43** | **94.04** | **68.46** |
> | **LoCoMo**        |           |           |           |           |           |           |           |           |
> | w/o Classify      | 20.75     | 20.75     | 36.56     | 30.86     | 44.06     | 34.16     | 53.93     | 37.31     |
> | w/ Classify       | **21.90** | **21.90** | **38.11** | **32.17** | **45.06** | **35.11** | **54.83** | **38.24** |
> | **LongMemEval-M** |           |           |           |           |           |           |           |           |
> | w/o Classify      | 33.19     | 33.19     | 57.87     | 41.04     | 68.09     | 46.46     | 82.98     | 51.77     |
> | w/ Classify       | **35.53** | **35.53** | **62.17** | **44.47** | **70.21** | **49.14** | **82.98** | **54.48** |
>
> As shown in Table 1, after introducing memory-type routing, **retrieval performance across all metrics consistently improved**, indicating that **queries were correctly routed to their corresponding memories**.  Moreover, for the more challenging recall@1 and recall@3 metrics, the improvement in retrieval precision is most significant, indicating that **the router effectively filters out some interfering memories through type-based filtering**.

---

> ### Author Response · Authors · 2025-11-21
> **Response to Reviewer Ajqg [Part 2-1]**
>
> ## **2. The Effectiveness of Type-Adaptive Retrieval Mechanisms**
>
> > **Q2**: Can the authors provide evidence or analysis showing that the observed performance improvement specifically arises from type-adaptive retrieval, rather than general architectural or training advantages?
> >
> > **W1**: the experiments only demonstrate that MemoType improves downstream performance, without directly showing that the type-adaptive retrieval mechanism itself is responsible for these gains.
>
> Thanks for the constructive feedback. We sincerely apologize for any lack in our initial submission. The impact of our proposed type-adaptive strategy on retrieval performance comes from two aspects: 1) **Only retrieving memories that correspond to the query type**, the performance impact resulting from filtering out partial memories; 2) **Applying differentiated strategies to different memory types**.   Below, we decouple these two parts and analyze their respective impacts on performance.
>
> - **Memory Filtering with Query Type**
>
>   **The first contribution of type-adaptive retrieval is filtering the memory corpus based on the query type**, which effectively excludes irrelevant memories. To evaluate the impact of this filtering, we designed two baseline settings:
>
>   - **"w/ Classify"**: Filters the memory corpus based on the query's category.
>   - **"w/o Classify"**: Retrieves from the entire memory corpus without category filtering.
>
>   No additional retrieval strategies were applied in these settings to ensure a fair comparison. The **ablation results in Table 1** (Table 1 is shown in the previous response) clearly show that applying query type classification consistently improves all retrieval metrics. These results validate that **type-based filtering effectively filters irrelevant memories, thereby enhancing retrieval performance**.
>
> - **Hybrid Retrieval Strategy**
>
>   **The second contribution lies in our hybrid retrieval strategy, which applies tailored strategies to different memory types**. We compared this **hybrid strategy** with **a single strategy** (all memory types employ the same retrieval-enhancing strategy) and investigated its impact on retrieval performance. As shown in Table 2, our hybrid retrieval strategy outperforms the best single-strategy baseline across all datasets.
>
>   These results confirm that the **hybrid retrieval strategy effectively leverages the unique advantages of different memory types to further boost performance**.
>
> **Table 2: The ablation study of the proposed hybrid strategy on three datasets.**
>
> | Data         | LongMemEval-S |           |           | LoCoMo    |           |           | LongMemEval-M |           |           |
>   | ------------ | ------------- | --------- | --------- | --------- | --------- | --------- | ------------- | --------- | --------- |
>   | Metric       | Recall@1      | Recall@3  | Recall@5  | Recall@1  | Recall@3  | Recall@5  | Recall@1      | Recall@3  | Recall@5  |
>   | Native       | 43.40         | 73.62     | 83.83     | 20.75     | 36.56     | 44.06     | 33.19         | 57.87     | 68.09     |
>   | Key          | 43.82         | 75.10     | 84.25     | 20.90     | 37.06     | 44.61     | 33.19         | 60.63     | 66.59     |
>   | Fake Memory  | 51.06         | 78.72     | 87.44     | 23.23     | 42.40     | 49.48     | 41.48         | 69.36     | 78.72     |
>   | Our Strategy | **55.74**     | **81.06** | **88.72** | **26.54** | **43.15** | **50.60** | **46.17**     | **71.91** | **80.21** |

---

> ### Author Response · Authors · 2025-11-21
> **Response to Reviewer Ajqg [Part 2-2]**
>
> - **Strategy Contributions of Each Type**
>
>   To investigate the **independent contribution of each memory strategy**, we supplemented the study with ablation experiments for each strategy category, as shown in Table 3. Each row of data represents the effect of applying only one strategy (GM, PM, or EM) of one specific memory type. **The experiment and related statement have been added in `Table 10` on Page 24 and  `Appendix D.1` on Page 22 in the revised version.**
>
>   The results in Table 3 show that **each memory type strategy independently contributes to performance improvement, with the EM strategy yielding the most significant gains**.
>
>   **Table 3: The ablation study of different retrieval strategies on three datasets.**
>
>   |                   | Recall@1          | NDCG@1            | Recall@3          | NDCG@3            | Recall@5          | NDCG@5            | Recall@10         | NDCG@10           |
>   | ----------------- | ----------------- | ----------------- | ----------------- | ----------------- | ----------------- | ----------------- | ----------------- | ----------------- |
>   | **LongMemEval-S** |                   |                   |                   |                   |                   |                   |                   |                   |
>   | Native            | 46.17             | 46.17             | 77.23             | 57.51             | 85.53             | 63.43             | 94.04             | 68.46             |
>   | with GM Strategy  | **47.45 (+1.28)** | **47.45 (+1.28)** | **77.87 (+0.64)** | **58.16 (+0.65)** | **85.53 (+0.00)** | **63.81 (+0.38)** | 93.92 (-0.12)     | **68.68 (+0.22)** |
>   | with PM Strategy  | **47.87 (+1.70)** | **47.87 (+1.70)** | **77.28 (+0.05)** | **58.26 (+0.75)** | **86.81 (+1.28)** | **64.71 (+1.28)** | **94.17 (+0.13)** | **69.49 (+1.03)** |
>   | with EM Strategy  | **52.77 (+6.60)** | **52.77 (+6.60)** | **80.28 (+3.05)** | **62.02 (+4.51)** | **87.45 (+1.92)** | **66.98 (+3.55)** | 93.92 (-0.12)     | **71.30 (+2.84)** |
>   | **LoCoMo**        |                   |                   |                   |                   |                   |                   |                   |                   |
>   | Native            | 21.9              | 21.9              | 38.11             | 32.17             | 45.06             | 35.11             | 54.83             | 38.24             |
>   | with GM Strategy  | **22.00 (+0.10)** | **22.00 (+0.10)** | **38.12 (+0.01)** | **32.22 (+0.05)** | **45.12 (+0.06)** | **35.15 (+0.04)** | **54.88 (+0.05)** | **38.30 (+0.06)** |
>   | with PM Strategy  | **23.51 (+1.61)** | **23.51 (+1.61)** | **40.68 (+2.57)** | **34.33 (+2.16)** | **47.89 (+2.83)** | **37.22 (+2.11)** | **57.00 (+2.17)** | **40.39 (+2.15)** |
>   | with EM Strategy  | **24.82 (+2.92)** | **24.82 (+2.92)** | **40.58 (+2.47)** | **34.57 (+2.40)** | **47.73 (+2.67)** | **37.64 (+2.53)** | **59.26 (+4.43)** | **41.37 (+3.13)** |
>   | **LongMemEval-M** |                   |                   |                   |                   |                   |                   |                   |                   |
>   | Native            | 35.53             | 35.53             | 62.17             | 44.47             | 70.21             | 49.14             | 82.98             | 54.48             |
>   | with GM Strategy  | **36.81 (+1.28)** | **36.81 (+1.28)** | **62.55 (+0.38)** | **44.75 (+0.28)** | **71.06 (+0.85)** | **49.66 (+0.52)** | **83.19 (+0.21)** | **54.80 (+0.32)** |
>   | with PM Strategy  | **37.02 (+1.49)** | **37.02 (+1.49)** | **64.17 (+2.30)** | **46.47 (+2.00)** | **73.40 (+3.19)** | **51.94 (+2.80)** | **84.26 (+1.28)** | **56.55 (+2.07)** |
>   | with EM Strategy  | **43.40 (+7.87)** | **43.40 (+7.87)** | **69.15 (+6.98)** | **50.67 (+6.20)** | **76.17 (+5.96)** | **55.04 (+5.90)** | **85.96 (+2.98)** | **59.91 (+5.43)** |

---

> ### Author Response · Authors · 2025-11-21
> **Response to Reviewer Ajqg [Part 3]**
>
> ## **3. Dataset Supplement**
>
> > **Q3**: Why were personalized or task-oriented benchmarks (e.g., PerLTQA, MEMTRACK) not included in the evaluation, given their relevance to memory categorization and realistic long-term interaction scenarios?
>
> Thanks for your valuable suggestion. We have supplemented corresponding retrieval and generation experiments on the PerLTQA dataset, as shown in Tables 4 and 5, to demonstrate the effectiveness of MemoType.  **The experiments have been added in `Table 2` on Page 7 and  `Table 3` on Page 8 in the revised version.**
>
> As shown in Table 4, **our method achieves the best retrieval performance across all metrics**, significantly outperforming existing baselines. Similarly, in **Table 5**, our approach demonstrates superior performance in question-answering tasks. These results validate the robustness of our method in handling complex task-oriented memory scenarios.
>
> **Table 4: The Retrieval Performance of MemoType on the PerLTQA dataset.**
>
> | Method    | Recall@1  | NDCG@1    | Recall@3  | NDCG@3    | Recall@5  | NDCG@5    | Recall@10 | NDCG@10   |
> | --------- | --------- | --------- | --------- | --------- | --------- | --------- | --------- | --------- |
> | HyDE      | 17.09     | 17.09     | 56.06     | 53.16     | 66.42     | 58.01     | 79.49     | 62.52     |
> | Mill      | 51.87     | 51.87     | 77.66     | 75.29     | 84.36     | 78.47     | 91.40     | 80.92     |
> | Query2Doc | 26.15     | 26.15     | 62.26     | 59.34     | 72.90     | 64.36     | 83.47     | 68.04     |
> | A-Mem     | 25.11     | 25.11     | 30.47     | 39.57     | 45.50     | 41.91     | 59.77     | 56.67     |
> | HippoRAG2 | 41.47     | 41.47     | 72.23     | 70.02     | 78.37     | 72.91     | 85.80     | 75.50     |
> | Ours      | **59.48** | **59.48** | **80.39** | **78.69** | **86.39** | **81.05** | **92.47** | **82.83** |
>
> **Table 5: The Question Answer Performance of MemoType on PerLTQA dataset.**
>
> | Method    | GPT4J     | F1        | BLEU      | Rouge1    | Rouge2    | RougeL    | RougeLsum | BERTScore |
> | --------- | --------- | --------- | --------- | --------- | --------- | --------- | --------- | --------- |
> | HyDE      | 39.39     | 30.80     | 7.21      | 32.59     | 16.41     | 26.82     | 26.90     | 88.86     |
> | Mill      | 51.37     | 36.65     | 10.61     | 38.41     | 21.40     | 32.44     | 32.50     | 89.85     |
> | Query2Doc | 42.32     | 31.95     | 7.82      | 33.72     | 17.46     | 27.95     | 28.01     | 89.06     |
> | A-Mem     | 32.12     | 34.96     | 5.20      | 29.95     | 19.64     | 31.94     | 22.01     | 87.74     |
> | HippoRAG2 | 49.26     | 34.44     | 8.89      | 36.24     | 20.01     | 30.48     | 30.57     | 89.42     |
> | Ours      | **52.85** | **42.49** | **17.17** | **44.32** | **26.49** | **38.69** | **38.66** | **90.92** |
>
> However, we must report that **the current version of MemoType cannot process datasets like MEMTRACK**. The primary reason is that **TriMEM datasets are constructed solely with dialogue memory annotations**, corresponding to declarative memory in cognitive psychology theory [1]. Declarative memory is a form of long-term memory that involves conscious recollection and includes memory for facts (semantic memory) and events (episodic memory).  However, **MEMTRACK primarily consists of tool logs and step records, which belong to the Procedural Memory of Nondeclarative Memory**. Procedural memory is memory that involves performing certain actions.
>
> Since **the primary focus of this paper is on dialogue memory** (i.e., declarative memory), classical works such as A-Mem [2], HippoRAG [3], and MemoryBank [4] have focused solely on dialogue memory—our constructed TriMEM dataset lacks relevant categories for nondeclarative memory. Furthermore, we did not design enhancement strategies for nondeclarative memory, making it challenging for models to effectively classify or enhance such memories.
> Nevertheless, we greatly appreciate your valuable feedback. In subsequent research, we will explore methods to bridge this gap, such as **introducing annotations for nondeclarative related memories and designing corresponding processing strategies**. Once again, thanks for your constructive suggestions, which will guide us in refining our research work.
>
> [1] Eysenck, Michael W., and Mark T. Keane. Cognitive psychology: A student's handbook. Psychology press, 2020.
>
> [2] Xu, Wujiang, et al. "A-mem: Agentic memory for llm agents." arXiv preprint arXiv:2502.12110 (2025).
>
> [3] Jimenez Gutierrez, Bernal, et al. "Hipporag: Neurobiologically inspired long-term memory for large language models." Advances in Neural Information Processing Systems 37 (2024): 59532-59569.
>
> [4] Zhong, Wanjun, et al. "Memorybank: Enhancing large language models with long-term memory." Proceedings of the AAAI Conference on Artificial Intelligence. Vol. 38. No. 17. 2024.

---

> ### Author Response · Authors · 2025-11-21
> **Response to Reviewer Ajqg [Part 4]**
>
> ## **4. The Generalization of Classifier**
>
> > **Q4**: How does the simple BERT-based classifier generalize beyond the TriMEM dataset? Have the authors tested it on out-of-domain data or with alternative encoders (e.g., DeBERTa, RoBERTa) to assess robustness?
>
> Thanks for your insightful question and valuable suggestions. To address your concerns and further validate the robustness of our method, we conducted **classification experiments on the PerLTQA dataset to evaluate the classifier's capability on out-of-domain data in Table 6**, and supplemented the **ablation experiments for the BERT encoder in Table 7**. **The experiment and related statement have been added in `Table 11` on Page 28 and  `Appendix D.2` on Page 23 in the revised version.**
>
> Specifically, we tested our classifier on the PerLTQA dataset to verify its performance beyond the TriMEM dataset. As shown in Table 6, the classifier achieved excellent results across all categories, with ACC and F1 scores reaching 100% for "Profile Description" and optimal scores for "Social Relationship" and "Events." These findings demonstrate **the strong generalization ability of our classifier**.
>
> **Table 6: The Classification Performance of the classifier on the PerLTQA dataset.**
>
> |      Model       | Profile Description |            | Social Relationship |           |  Events   |           |
> | :--------------: | :-----------------: | :--------: | :-----------------: | :-------: | :-------: | :-------: |
> |                  |         ACC         |     F1     |         ACC         |    F1     |    ACC    |    F1     |
> |     Qwen3-8b     |        88.65        |   93.98    |        57.21        |   72.78   |   46.23   |   63.23   |
> | Gemini-2.5-Flash |       100.00        |   100.00   |        75.06        |   85.75   |   91.83   |   95.74   |
> |   GPT-4o-Mini    |       100.00        |   100.00   |        89.47        |   94.44   |   92.00   |   95.83   |
> |    Qwen3-32b     |       100.00        |   100.00   |        96.64        |   98.29   |   87.55   |   93.36   |
> |       Ours       |     **100.00**      | **100.00** |      **97.16**      | **98.87** | **95.70** | **98.85** |
>
> In addition, we performed ablation studies to evaluate the impact of using alternative encoders, such as RoBERTa and DeBERTa. As evidenced in Table 7, both RoBERTa and DeBERTa achieved competitive results after fine-tuning on the TriMEM dataset, indicating that **our dataset is not restricted to a specific model architecture**. Notably, RoBERTa delivered better classification performance compared to the baseline BERT model, highlighting its superior modeling capacity.
>
> **Table 7: The Ablation Study of BERT Encoder.**
>
> |   LongMemEval-S   |   Recall@1   |   NDCG@1   |   Recall@3   |   NDCG@3   |   Recall@5   |   NDCG@5   |   Recall@10   |   NDCG@10   |
> | :---------------: | :----------: | :--------: | :----------: | :--------: | :----------: | :--------: | :-----------: | :---------: |
> |       BERT        |    55.74     |   55.74    |    81.06     |   63.40    |    88.72     |   68.64    |     93.62     |    72.54    |
> |      DeBERTa      |    55.32     |   55.32    |    81.91     |   63.19    |    89.36     |   68.90    |     95.74     |    72.62    |
> |      RoBERTa      |    58.09     |   58.09    |    83.83     |   65.27    |    90.21     |   70.70    |     95.53     |    74.41    |
> |    **LoCoMo**     | **Recall@1** | **NDCG@1** | **Recall@3** | **NDCG@3** | **Recall@5** | **NDCG@5** | **Recall@10** | **NDCG@10** |
> |       BERT        |    26.54     |   26.54    |    43.15     |   36.77    |    50.60     |   39.79    |     61.48     |    43.56    |
> |      DeBERTa      |    26.74     |   26.74    |    44.16     |   38.08    |    51.71     |   41.15    |     62.08     |    44.56    |
> |      RoBERTa      |    27.49     |   27.49    |    43.96     |   37.46    |    51.31     |   40.52    |     62.13     |    44.27    |
> | **LongMemEval-M** | **Recall@1** | **NDCG@1** | **Recall@3** | **NDCG@3** | **Recall@5** | **NDCG@5** | **Recall@10** | **NDCG@10** |
> |       BERT        |    46.17     |   46.17    |    71.91     |   52.95    |    80.21     |   58.35    |     87.45     |    62.29    |
> |      DeBERTa      |    46.38     |   46.38    |    68.94     |   49.73    |    77.66     |   55.55    |     86.81     |    59.91    |
> |      RoBERTa      |    49.15     |   49.15    |    69.57     |   51.96    |    79.79     |   57.52    |     88.51     |    62.12    |
>
> We sincerely appreciate your suggestion, which has helped us explore additional directions to improve the performance and generalizability of our method.

---

> ### Author Response · Authors · 2025-11-21
> **Response to Reviewer Ajqg [Part 5]**
>
> ## **5. Evaluation Protocol**
>
> > **Q5**: The paper reports outperforming much larger models (e.g., GPT-4o-Mini, Qwen3-32B). Could the authors clarify the evaluation protocol and whether there are benchmark-specific advantages or data overlaps that might explain this result?
>
> Thank you for your valuable feedback. We apologize for not making the experimental setup clear in the paper. The classification experiments in Table 9 of the appendix **can't demonstrate that the BERT model outperforms models like GPT-4o-mini and Qwen3-32B**. Since the **BERT model here is a fine-tuned version** trained on labeled datasets, while the other models compared **use prompts** to classify text. The experiment in Table 9 is designed to demonstrate that **existing prompt-based classification strategies cannot achieve precise classification of dialogue**.
>
> In fact, during the early stages of our method development, we experimented with using LoRA fine-tuning on Llama-3.1-8B-Instruct for memory classification. The preliminary results showed that **LoRA-fine-tuned 8B models performed worse than the fine-tuned BERT model in classification tasks**.
>
> For these experiments, we labeled approximately 3,000 samples from the LongMemEval-S dataset to fine-tune both Llama-3.1-8B-Instruct and BERT. Since LongMemEval-S does not provide memory labels, we evaluated classification accuracy based on its impact on retrieval performance (misclassifications degrade retrieval results). The preliminary results are shown in the table below, where **BERT consistently outperforms Llama-3.1-8B-Instruct across various retrievers**.
>
> **Table 8: The Retrieval Performance Comparison with Different Router Models**
>
> |              Model               | Recall@1  | Recall@3  | Recall@5  | Recall@10 |
> | :------------------------------: | :-------: | :-------: | :-------: | :-------: |
> |       **Retriever: BM25**        |           |           |           |           |
> |              Native              |   44.46   |   61.70   |   68.93   |   75.74   |
> |               BERT               | **46.81** | **65.53** | **71.91** | **79.57** |
> | Llama-3.1-8B-Instruct (Lora+DPO) |   45.11   |   61.91   |   69.36   |   76.81   |
> | Llama-3.1-8B-Instruct (Lora+SFT) |   45.32   |   64.47   |   69.57   |   77.66   |
> |    **Retriever: Contriever**     |           |           |           |           |
> |              Native              |   43.40   |   73.61   |   83.82   |   94.25   |
> |               BERT               | **48.30** | **77.66** | **86.17** | **94.26** |
> | Llama-3.1-8B-Instruct (Lora+DPO) |   43.62   |   73.83   |   83.83   |   94.47   |
> | Llama-3.1-8B-Instruct (Lora+SFT) |   44.89   |   73.40   |   82.34   |   90.43   |
> |       **Retriever: MPNet**       |           |           |           |           |
> |              Native              |   26.38   |   50.85   |   63.62   |   80.00   |
> |               BERT               | **34.47** | **61.06** | **73.19** | **84.68** |
> | Llama-3.1-8B-Instruct (Lora+DPO) |   27.02   |   51.70   |   64.47   |   80.21   |
> | Llama-3.1-8B-Instruct (Lora+SFT) |   27.45   |   54.47   |   65.74   |   80.21   |
>
> > *We hope these responses address your concerns comprehensively. Feel free to let us know if you have any additional questions, and we are more than happy to provide further clarification on any aspect of our work.*
> >
> > *Respectfully and sincerely,*
> >
> > *The Authors*

---

### Official Review · Reviewer_jXah · 2025-11-01

**Soundness:** 2
**Presentation:** 2
**Contribution:** 2
**Rating:** 2
**Confidence:** 4

**Summary:**

To address the challenge that, given the topic-rich, scenario-complex, and boundary-blurred nature of memory scenarios, achieving precise classification of memories is not easy, this paper proposes a memory multi-class benchmark in this paper, termed TriMEM. TriMEM comprises 6,000 dialogue samples, providing precise annotations for memory types across diverse topics and scenarios. Building upon this foundation, this work proposes a memory framework, named MemoType.

**Strengths:**

1. This paper proposes a memory multi-class benchmark.
2. This work proposes a memory augmentation framework to classify and use memory for QA.
3. The paper is well-structured.

**Weaknesses:**

1. As far as I know, ref[1] has presented a benchmark for multi-class memory. However, this paper did not describe the core difference from [1].
2. This paper includes three categories in the data. Why these three categories? Do we need other categories?
3. The introduction of memory types is insufficient. The authors should provide more details about the types of memory.


[1] Du et al., Perltqa: A personal long-term memory dataset for memory classification, retrieval, and fusion in question answering. In Proceedings of the 10th SIGHAN Workshop on Chinese Language Processing (SIGHAN-10), pp. 152–164, 2024.

**Questions:**

1. This paper includes three categories in the data. Why these three categories? Do we need other categories?
2. What is the core difference between this work and [1]?
3. How can the MemoType framework balance the importance among different types of memory?

[1] Du et al., Perltqa: A personal long-term memory dataset for memory classification, retrieval, and fusion in question answering. In Proceedings of the 10th SIGHAN Workshop on Chinese Language Processing (SIGHAN-10), pp. 152–164, 2024.

---

> ### Author Response · Authors · 2025-11-21
> **Response to Reviewer jXah [Part 1-1]**
>
> **Dear Reviewer jXah:**
>
> Thank you so much for your thoughtful review and valuable feedback. We sincerely appreciate the time and effort you devoted to evaluating our work. Below, we have provided detailed responses to your comments.
>
>
> ## **1. The Core Difference between MemoType with PerLTQA[1].**
>
> > **W1**: As far as I know, ref[1] has presented a benchmark for multi-class memory. However, this paper did not describe the core difference from [1].
> >
> > **Q2**: What is the core difference between this work and [1]?
>
> Thanks for the comments. We apologize for any confusion caused by the paper. We will elaborate on the differences between MemoType and PerLTQA [1] from both **Methodological and Data Perspectives**. **The statement has been added in `Section 2.2` on Page 3 in the revised version.**
>
> ### **1. 1 Methodological Perspective**
>
> PerLTQA is a benchmark paper whose primary contribution is to propose a Question-Answering and Classification benchmark, **without proposing specific tailored strategies for different memory types**.
>
> In contrast, **MemoType** introduces two key methodological innovations:
>
> - **Memory and Query Routing**: With the guidance of the TriMEM dataset, MemoType finetunes a router model to adaptively route memory. As for the query routing, MemoType leverages the extensive knowledge of large language models (LLMs) to generate multiple "fake memories" for each query and employs a voting mechanism to achieve precise classification of queries. With the memory and query routing, MemoType can retrieve the memory with corresponding query types rather than retrieving the whole memory corpus, thereby enhancing the retrieval efficiency.
> - **Type-specific Retrieval Mechanisms**: MemoType designs distinct retrieval and storage mechanisms tailored to each memory type. For episodic memory, we employ an event element matching method that decouples time, person, and location. For personal semantic memory, we use synonym expansion to bridge the semantic gap. For general semantic memory, we enhance retrieval by generating supplementary "fake memories" based on LLM-embedded knowledge.
>
> **To the best of our knowledge, MemoType is the first memory framework that adaptively classes dialogue memory and designs distinct retrieval and storage mechanisms for different memory types.**
>
> [1] Du, Yiming, et al. "PerLTQA: A personal long-term memory dataset for memory classification, retrieval, and fusion in question answering." *Proceedings of the 10th SIGHAN Workshop on Chinese Language Processing (SIGHAN-10)*. 2024.

---

> ### Author Response · Authors · 2025-11-21
> **Response to Reviewer jXah [Part 1-2]**
>
> ### **1.2 Data Perspective**
>
> Despite being a pioneering dataset in memory classification, the **PerLTQA dataset [1] still suffers from the following drawbacks**:
>
> - **Single Memory Type.**  PerLTQA first determines the memory category before generating the corresponding memory, which leads to a problem: **the generated memory only has a single label**. In real-world scenarios, a single conversation may involve multiple topics, resulting in multiple memory categories. Single-label memories fail to reflect real-world scenarios and artificially reduce classification difficulty. In the proposed TriMEM dataset, we constructed memories from real scenarios first, then annotated and refined three different memory types labels, which avoids the single memory type issue.
>
> - **Incomplete Memory Type.** PerLTQA [1] predefines four information types (personal profiles, social relationships, events, dialogues). Among these, personal profiles and social relationships are categorized as semantic memory, while events and dialogues are categorized as episodic memory. However, such classification doesn't cover all memories:
>
>   **i. Incomplete Semantic Memory.** The semantic memory in PerLTQA consists solely of personal semantic memory, **neglecting general semantic memory**. Referring to the cognitive psychology [2] definition of semantic memory: "A form of long-term memory consisting of general knowledge about the world, concepts, language, and so on" (Definition on Page 285 in [2]), general world knowledge and other memories unrelated to the user also belong to the semantic memory domain.
>
>   **ii. Incomplete Episodic Memory.**The episodic memory in PerLTQA focuses solely on past specific events, representing retrospective episodic memory, while **lacking prospective episodic memory** for future planning (Definition on Page 367 in [2] ).
>
>   In the constructed TriMEM dataset, referring to the memory classification principle in cognitive psychology [2], we comprehensively consider different memory categories. More detailed classification can be found in Table 1 in our paper.
>
> - **Coarse-Grained Memory Type.** **PerLTQA uniformly classifies all dialogues as episodic memory**. Such coarse-grained categorization is unreasonable. Dialogues are crucial retrieval objects in existing memory scenarios. For instance, in the two common memory benchmarks LoCoMo [3] and LongMemEval [4], **dialogues are the primary and exclusive retrieval objects**. Moreover, in real-world scenarios, accessible data typically exists in dialogue form, while user profiles and social relationships are often difficult to obtain. Therefore, **the distinction between user profiles and dialogues as separate types lacks transferability in practical memory scenarios**.
>
> **To the best of our knowledge, TriMEM is the first dialogue memory multi-classification dataset.**
>
> [1] Du, Yiming, et al. "PerLTQA: A personal long-term memory dataset for memory classification, retrieval, and fusion in question answering." *Proceedings of the 10th SIGHAN Workshop on Chinese Language Processing (SIGHAN-10)*. 2024.
>
> [2]  Eysenck, Michael W., and Mark T. Keane. *Cognitive psychology: A student's handbook*. Psychology press, 2020.
>
> [3] Maharana A, Lee D H, Tulyakov S, et al. Evaluating Very Long-Term Conversational Memory of LLM Agents[C]//Proceedings of the 62nd Annual Meeting of the Association for Computational Linguistics (Volume 1: Long Papers). 2024: 13851-13870.
>
> [4] Wu D, Wang H, Yu W, et al. LongMemEval: Benchmarking Chat Assistants on Long-Term Interactive Memory[C]//The Thirteenth International Conference on Learning Representations.

---

> ### Author Response · Authors · 2025-11-21
> **Response to Reviewer jXah [Part 2]**
>
> ## **2. The Reason for Selecting the Three Categories.**
>
> > **W2 & Q1**: This paper includes three categories in the data. Why these three categories? Do we need other categories?
>
> Thanks for the comments. Below, we will clarify the rationale for choosing these categories. **The statement has been added in `Appendix A.1` on Page 14 in the revised version.**
>
> ### **2.1 The Completeness of Types**
>
> - Our classification is based on well-established research in **cognitive psychology [2] on declarative memory**. **Declarative memory** is a form of long-term memory that involves conscious recollection and **includes memory for facts (semantic memory) and events (episodic memory)**, which often refers to memories that can be "declared" or described (Dialogue Memory).   This classification comprehensively covers all aspects of dialogue memory, as illustrated in [Figure 7.2](https://anonymous.4open.science/r/MemoType-5578/Fig/memory_class.png) on [2].
> - For Nondeclarative memory, such as procedural memory or priming memory, etc., it falls outside the scope of this study (dialogue memory) because it cannot be expressed linguistically.  Therefore, **the proposed three memory types encompass all dialogue memory, with no need for additional types**.
>
> ### **2.2 The Necessity of Categorizing Semantic Memory**
>
> In terms of semantic memory, we categorize it into **Personal Semantic Memory** and **General Semantic Memory** based on two key considerations:
>
> - **Category Balance.** Due to the relatively low proportion of events in daily conversations, **categorizing dialogue into semantic memory and episodic memory leads to the significant category imbalance problem**. For example, in the whole LongMemEval-S dataset, **semantic memory accounts for approximately 87.64%**, while **episodic memory constitutes only 12.35%**. Such an imbalance causes the model to prioritize the larger category during optimization, while evaluation results may be dominated by the larger category, thereby failing to accurately reflect the model's overall capability.
> - **Semantic Gap Between Categories.** To analyze the semantic gap between General Semantic memory (**GM**) and Personal Semantic Memory (**PM**), we plotted a TSNE figure representing these two memory types under the Contriever retriever on sampled LongMemEval-S data. As shown in the **[TSNE Visualization Figure](https://anonymous.4open.science/r/MemoType-5578/Fig/LongMemEval-S_TSNE.pdf)**, both categories exhibit **distinct clustering patterns**, indicating significant semantic differences between personal and general semantic memory. Additionally, we supplemented **[GM Word Cloud Visualizations](https://anonymous.4open.science/r/MemoType-5578/Fig/General_Semanic_Memory_Wordcloud.pdf)** and **[PM Word Cloud Visualizations](https://anonymous.4open.science/r/MemoType-5578/Fig/Personal_Semanic_Memory_Wordcloud.pdf)**, revealing that General Semantic Memory emphasizes common terms like "assistant" and "offer," while Personal Semantic Memory focuses on user-specific terminology such as "planning "and "user." This further highlights the distinctions between the two memory types.
>
> ## **3. The Supplementary Induction of Memory Types**
>
> > **W3**: The introduction of memory types is insufficient. The authors should provide more details about the types of memory.
>
> Thanks for the feedback. We apologize for any confusion caused by the paper.  **The statement has been added in `Appendix A.2` on Page 14 in the revised version.**
>
> - **Episodic Memory**
>
>   Episodic memory refers to memories of specific events or occurrences tied to contextual details such as time, location, participants, and activities. These memories can include both past experiences and plans for the future. For example:
>
>   "Last summer, I attended a three-day conference in Paris where I presented my research on renewable energy."
>
>   "Next Friday, I’m scheduled to meet my supervisor at 10 a.m. to review the final draft of my thesis."
>
> - **Personal Semantic Memory**
>
>   Personal Semantic Memory refers to long-term knowledge related to an individual's identity, preferences, habits, or intentions. Unlike Episodic Memory, it focuses on general personal information rather than specific events. For instance:
>
>   "I have been working as a data scientist for a long time, and I specialize in natural language processing."
>
>   "I usually prefer hiking in the mountains over beach vacations because I enjoy the tranquility of nature."
>
> - **General Semantic Memory**
>
>   General Semantic Memory encompasses universally applicable knowledge, objective descriptions, or facts about the world. It is not tied to any specific individual or event. Examples include:
>
>   "The Great Wall of China is over 13,000 miles long and was constructed to protect against invasions."
>
> [2] Eysenck, Michael W., and Mark T. Keane. Cognitive psychology: A student's handbook. Psychology press, 2020.

---

> ### Author Response · Authors · 2025-11-21
> **Response to Reviewer jXah [Part 3]**
>
> ## **4. The Balance among Different Memory Types**
>
> > **Q3**: How can the MemoType framework balance the importance among different types of memory?
>
> Thanks for the valuable feedback. **Our method does not prioritize one memory type over another**. Instead, the framework **dynamically determines each query type and applies a tailored retrieval strategy**. This ensures that each memory type is treated equally but with strategies optimized for its characteristics.
>
> - **Memory Type Router**
>
>   To classify the memory type, we use a trained router model, which categorizes memories into three types. The router achieves precise classification through a multi-label task. By doing so, the model **adaptively identifies the memory type without imposing an importance hierarchy**.
>
> - **Query Type Router**
>
>   For each query, we leverage a Large Language Model (LLM) to generate five fake memory candidates. The memory type that occurs most frequently among the fake memories is defined as the query type (majority voting mechanism). Types appearing in false memories will be identified as the retrieval scope for the query. **This adaptive process can determine the query's most possible type** while keeping the retrieval scope confined to relevant memory types, thereby reducing computational overhead and improving accuracy.
>
> - **Type-Specific Retrieval Strategies**
>
>   **Once the query's type is identified, the MemoType framework applies a retrieval strategy tailored to that particular type**. Episodic Memory: Focusing on event-specific elements like time, participants, and location. Personal Semantic Memory: The query is enhanced by generating additional keywords using the LLM for better retrieval. General Semantic Memory: Using fake memories to enhance retrieval.
>
> By adapting the retrieval process to the query's memory type, we ensure that no single memory type is inherently more important. Instead, the MemoType framework dynamically determines each query's type based on its content.
>
> > *We hope these responses address your concerns comprehensively. Feel free to let us know if you have any additional questions, and we are more than happy to provide further clarification on any aspect of our work.*
> >
> > *Respectfully and sincerely,*
> >
> > *The Authors*

---

### Meta-Review · Area_Chair_Kpts · 2026-01-08

**Summary:**

The paper proposes "MemoType," a framework that categorizes long-term memory in LLMs into three types to apply tailored retrieval strategies. The authors also introduce "TriMEM," a benchmark dataset with 6,000 annotated dialogue samples. While the reviewers acknowledged the intuitive appeal of categorizing memory and the effort put into the TriMEM benchmark, the consensus leans towards rejection due to significant concerns regarding novelty, generalization, and methodological depth.

**Reviewer Concerns:**

The most critical issue is the similarity to prior work.

Concerns about the generalization of the memory-type classifier.

Limitation of excluding procedural memory.

Given the incremental gains over strong baselines and the overlap with existing literature, the complexity introduced by maintaining distinct pipelines for strictly defined memory types may not be justified.

**Reviewer Scores:**

After the discussion, this is a borderline submission.

---

### Decision · Program_Chairs · 2026-01-26

Reject